# Structure and dynamics of Toll immunoreceptor activation in the mosquito Aedes aegypti

Yoann Saucereau [1], Thomas H. Wilson[1], Matthew C. K. Tang [1], Martin C. Moncrieffe[1], Steven W. Hardwick [1], Dimitri Y. Chirgadze [1], Sandro G. Soares [1], Maria Jose Marcaida [2], Nicholas J. Gay[1] & Monique Gangloff [1] ✉

*Aedes aegypti* has evolved to become an efficient vector for arboviruses but the mechanisms of host-pathogen tolerance are unknown. Immunoreceptor Toll and its ligand Spaetzle have undergone duplication which may allow neo-functionalization and adaptation. Here we present cryo-EM structures and biophysical characterisation of low affinity Toll5A complexes that display transient but specific interactions with Spaetzle1C, forming asymmetric complexes, with only one ligand clearly resolved. Loop structures of Spaetzle1C and Toll5A intercalate, temporarily bridging the receptor C-termini to promote signalling. By contrast unbound receptors form head-to-head homo-dimers that keep the juxtamembrane regions far apart in an inactive conformation. Interestingly the transcriptional signature of Spaetzle1C differs from other Spaetzle cytokines and controls genes involved in innate immunity, metabolism and tissue regeneration. Taken together our results explain how upregulation of Spaetzle1C in the midgut and Toll5A in the salivary gland shape the concomitant immune response.

Insect Toll receptors perform critical functions in both embryogenesis and innate immunity. They are part of an ancient defence system that also includes the immunodeficiency (IMD) pathway and are conserved in vertebrates as the Toll-like receptors. Tolls and TLRs have a modular structure with an ectodomain made up of leucine-rich repeats (LRRs) and associated capping structures, a single transmembrane helix and a cytosolic Toll/Interleukin 1 receptor (TIR) domains[1]. Most studies of innate immune function in insects have focussed on the fruit fly *Drosophila melanogaster*[2]. The *Drosophila* genome encodes 9 Toll receptors that have diverse roles in development and other areas of cellular regulation as well as in immunity. Immune function is mediated mainly by the Toll receptor 1 (Toll1), while Toll6 and Toll7 are expressed in the nervous system and have equivalent functions to vertebrate neurotrophin receptors[3].

Toll receptors are activated by a complementary family of six cytokine-like molecules called Spaetzle (Spz)[4]. Spz1 proteins are secreted as larger dimeric precursors consisting of a natively unstructured pro-domain, which is proteolytically cleaved upon activation and a C-terminal cystine-knot fold similar to that found in human neurotrophins, such as nerve growth factor[5,6]. Lys-type peptidoglycan from Gram-positive bacteria is detected by recognition proteins PGRP-SA and GNBP1 and these complexes activate a cascade of serine proteases[7]. The terminal proteinase, Spaetzle processing enzyme (SPE), has trypsin-like specificity and cleaves the Spz precursor specifically at the junction between the pro-domain and cystine-knot, which remain associated by non-covalent interactions[8]. In contrast Spz2 (also called neurotrophin-1, NT1) and Spz5 (NT2) are cleaved during secretion by furin-like proteases[9] and Spz5 is secreted in a mature form without its pro-domain. Growing evidence suggests

[1]Department of Biochemistry, University of Cambridge, 80 Tennis Court Road, Cambridge CB2 1GA, UK. [2]Institute of Bioengineering, School of Life Sciences, Ecole Polytechnique Fédérale de Lausanne, Lausanne, Switzerland. ✉e-mail: mg308@cam.ac.uk

promiscuity in ligand binding, with Toll1 and Toll7 recognising Spz1, Spz2 and Spz5 and, more controversially, vesicular stomatitis virus (VSV) virions[10–13], while Toll6 senses Spz2, Spz5 and possibly dsRNA[9,14].

Mosquitoes and fruit flies are both dipterans but they diverged in evolution about 260 million years ago[15,16]. Comparative genomic analysis of *Aedes aegypti*, *Anopheles gambiae* and *Drosophila melanogaster* reveals that most genes involved in innate immunity are conserved in the mosquito, including the Toll and Spaetzle families[17]. Nevertheless, the Toll family has undergone significant diversification with the loss of orthologs in mosquitoes (no Toll2 or Toll3 in *A. aegypti*, for example) and species-specific expansion of two more (Toll10 and Toll11). The underlying driving force of gene duplication is likely interconnected with the mosquito's change to a hematophagous diet and the evolutionary arms-race between pathogens and insects. In *Drosophila* Toll1 mediates most immune functions, while in *Aedes* two gene reduplications have occurred to produce an orthologous group of four closely related receptors Toll1A, Toll1B, Toll5A and Toll5B. The phylogeny does not provide unequivocal evidence as to which of these receptors function in immunity or if they have acquired new functions upon diversification.

Although there is a clear evolutionary relationship between insect Tolls and vertebrate TLRs the latter appear to have evolved from a common ancestor of insect Toll9 and adapted to directly recognise a diverse range of pathogen-associated molecular patterns (PAMPs) such a bacterial lipopolysaccharide[18]. Thus, vertebrate TLRs are bona fide pattern recognition receptors but insect Tolls are activated by endogenous cytokines or growth factors. Another distinction between Toll and TLRs is the kinetics of receptor activation. TLRs display positive cooperativity being activated over a narrow range of ligand concentrations[19]. By contrast, the function of Toll1 in development requires a diffused gradient of active Spaetzle with different threshold concentrations specifying cell fates along the embryonic dorso-ventral axis. The same is thought to occur in the patterning of immune responses.

Structural studies have provided insight into the molecular mechanism by which PAMPS activate the TLRs[20]. A common feature is stimulus-induced dimerization of the receptor ectodomains or alternatively conformational rearrangement of a preformed inactive dimer. For example, double-stranded RNA crosslinks two TLR3 ectodomains causing the juxtamembrane regions to move into close proximity and promote dimerization of the TIR domains in the cytosol[21]. However, TLR8 is expressed as a pre-formed dimer and the binding of small drug molecules or single-stranded RNA induces a conformational rearrangement of the dimer interface that results in the tilting of the two ectodomains and, like TLR3, moves the juxtamembrane C-termini into close proximity[22]. Similar mechanisms are likely operative in the activation of insect Tolls by Spz ligands. Biochemical evidence suggests that in insects the active complex has a stoichiometry of two molecules of receptor and two molecules of Spaetzle-C106 (2:2 complex)[23]. However, to date the only crystal structures of the Toll–C106 complex reveal a 1:1 complex with a binding mode that is reminiscent of mammalian neurotrophins[24,25]. The covalent cystine-knot dimer forms asymmetric contacts at the concave side of the N-terminal domain within the first ten LRRs. In contrast to TLRs, Spz does not induce dimerization of the receptor in the crystal structure possibly because negative cooperativity requires that the active signalling complex is unstable.

Functionally, the Toll pathway has been involved in resistance against the entomopathogenic fungus *Beauveria bassiana*[26] and Dengue viruses[27,28], with fungal infections limiting viral replication[29]. Differential expression of duplicated Toll5A and Spz1C suggests that this pathway is regulated in a tissue and microbe-specific manner[30,31]. Here, we use purified proteins to further characterise Toll5A-Spz1C interactions and the impact of their relative concentrations on stoichiometry. Single particle cryo-EM reveals that the receptor is reversibly assembled in an asymmetric complex, in which C-terminal regions are brought into proximity upon Spz1C binding, which is a structural requirement for signal transduction. Along with Spz1C-induced *A. aegypti* Aag2 cell signalling, our study sheds light on the effect of duplication and shows the importance of revisiting Toll signalling in hematophagous insects that impact human health.

## Results

### Cryo-EM structures of Toll5A and Spz1C

Recombinant Toll5A ectodomain was mixed with a 3-fold molar excess of Spz1C and the resulting complex was then purified by size exclusion chromatography immediately prior to grid preparation as described in Methods. Three oligomeric states of the receptor were observed at near-atomic resolutions, suggesting that Toll5A is highly dynamic in the presence of Spz1C. Homodimers of Toll5A at 3.41 Å maximum resolution had 85,810 particles, 2:1 heterodimers of Toll5A bound to Spz1C at 4.23 Å had 40,153 particles, and a 3:1 heterotrimer at 3.74 Å had 42,866 particles (Fig. 1 and Supplementary Fig. 1). While all three types of particles spread randomly across the grid, according to their large Euler angular distribution, they display significant variations in local resolutions, likely a consequence of their stoichiometric and conformational heterogeneity. Overall, the N-terminal LRR domains of each receptor chain were better resolved than their C-terminal moieties with the latter partially truncated upon density modification (Supplementary Fig. 2). Similarly, two ligands are observed but only one is well resolved, while the density of the second one is partial and seems to be restricted to the cys-knot core of the molecule. Although there is space to accommodate a second ligand without any steric clashes, the lack of resolution in this area does not allow visualisation. The heterodimer complex is therefore treated as a 2:1 complex. The vacated ligand binding site is occupied by another receptor chain in the 3:1 heterotrimer, illustrating the transition between homo- and heterodimerisation (Supplementary Fig. 1). The three Toll5A molecules observed are referred to as chain A (teal), chain B (dark blue) and chain C (powder blue) in superimposable conformations in all three structures (RMSD below 2 Å). The Spz1C homodimer is composed of chain D (for distal, in yellow) and chain P (for proximal, in orange), attached to Toll5A chain A at the concave side and to chain B at its "back" or convex side.

The structure of Toll5A homodimers (chains A and B) was solved at the highest overall resolutions (3.4–4.4 Å), with a head-to-head arrangement, maintaining the C-terminal juxtamembrane regions far apart (Fig. 1a and Supplementary Table 1). The Cα atoms of Cys-783 at the C-termini of each receptor chain are separated by 207 Å. Such a structural arrangement –if sterically possible when the receptor is expressed on the same cell– would ensure that Toll5A is locked in an inactivate state preventing TIR domain association and signalling. If receptors are situated on neighbouring cells, such an orientation might be relevant to cell adhesion.

The overall structure of Toll5A is comparable to DmToll with a conserved number of leucine-rich repeats in both the N- and C-terminal domains, and conserved cysteine-rich capping structures surrounding these domains. The sequences of Toll5A and DmToll1 are 30 % identical and 50 % similar. In particular, the extended N-terminal cap formed by two hairpins and a parallel β-sheet, which is involved in Spz binding, is conserved in *A. aegypti* Toll5A but not in *Drosophila* Toll-5, also known as Tehao, which remains an orphan receptor[32].

Superposition of Toll5A and DmToll structures reveals that the diameter of the N-terminal LRR solenoid of Toll5A is 5–10 Å greater compared to the *Drosophila* Toll1 receptor, with an inner diameter of about 50 Å and an outer diameter of 90 Å and an overall RMSD of 3 Å (Supplementary Fig. 3). Toll5A has nine predicted glycosylation sites, 6 of which were observed (Asn-linked glycans visible at positions 151, 194, 481, 521, 634 and 687) and found to restrict access to its surface. In

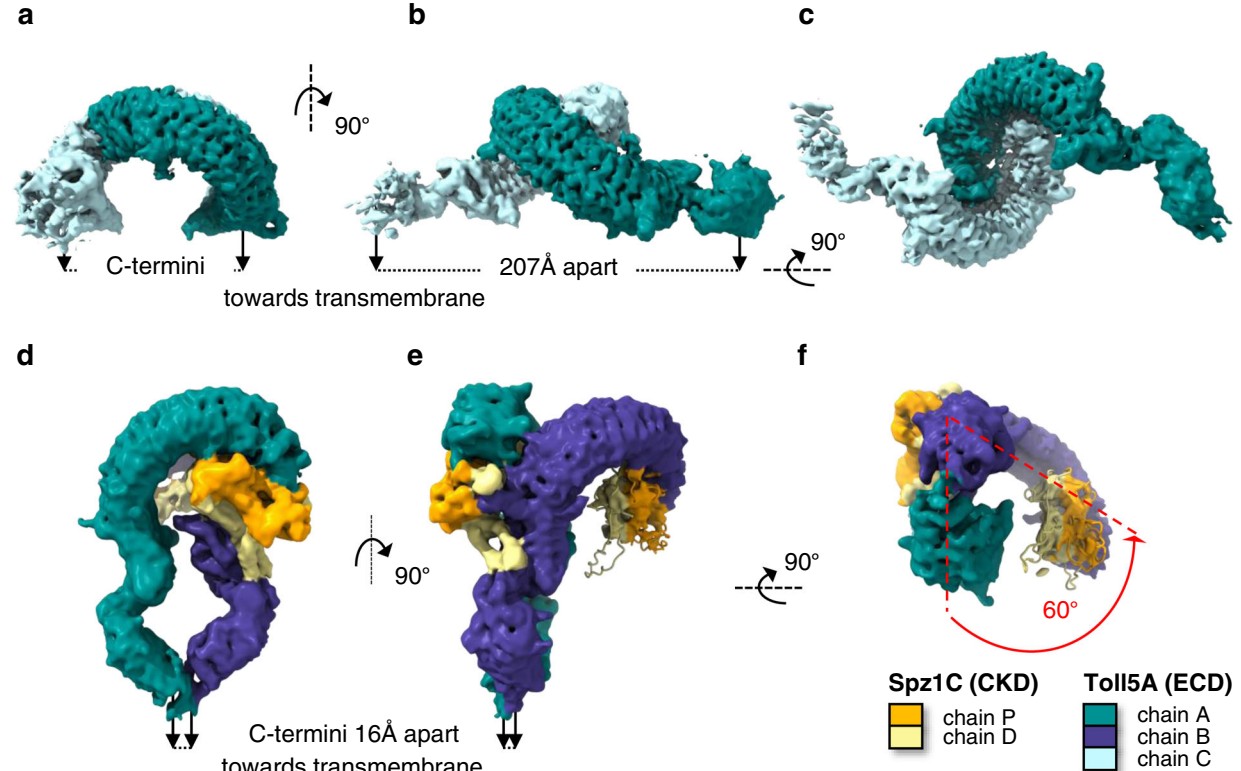

**Fig. 1 | Cryo-EM structures of Toll5A homodimer and Spz1C-ligated hetero-dimer.** Orthogonal and bottom-up views of cryo-EM density maps. Density is colour-coded according to protein content. **a**–**c** Toll5A homodimer viewed in three different orientations. **d**–**f** Spz1C-bound heterodimer. **f** View from the membrane up to emphasize the asymmetry of the complex with a 60° deviation of chain B compared to chain A. The density of the second ligand is severely truncated compared to the first one probably due to lack of contact at the dimerization interface.

contrast, DmToll has thirteen sites, which are not conserved with Toll5A, except for Asn-481 on the ascending flank of LRR15. Glycans restrict protein-protein contacts on the concave surface and the flanks of the receptor.

Toll5A forms head-head symmetric homodimers with LRRNT1 cap residues between Thr-34 and Tyr-79 binding the concave surface in the vicinity of Tyr-202 (LRR4) to Asn-419 (LRR13). For instance, Tyr-56 is hydrogen bonded to Tyr 226 for each chain (Fig. 2). The buried surface area is about 1000 Å². Interestingly, receptor-receptor contacts in Toll5A differ from those observed in *Drosophila* Toll structures[25,33], which involve the N-terminal capping region and the hinge region between LRR domains. The Toll5A homodimer interface overlaps with the Spz1C ligand binding site, which suggests a direct competition between receptor-receptor and receptor-ligand interactions. Moreover, the presence of unliganded receptor in the Toll5A-Spz1C sample suggests that ligand binding is reversible.

**Spz1C binding breaks receptor symmetry**

We observed a ligated heterodimer of Toll5A with a stoichiometry of two receptor ectodomains to one Spz1C ligand (2:1 complex) at a lower resolution (4.2–8.4 Å). Spz1C is a covalent dimer stabilised by two intermolecular disulphide bonds instead of one in *Drosophila* (Supplementary Fig. 3). The central disulphide bond between Cys-94 is conserved in DmSpz, while Spz1C has an additional intermolecular bond between Cys-59 located basally in an area directly involved in binding to the N-terminal concave surface of the receptor. Proximal Cys-59 is located within hydrogen bonding distance of the hydroxyl group of Toll5A Tyr-226 in LRR5 (Fig. 2). The concave interface formed by the N-terminus up to LRR7 buries only ~ 890 Å² of accessible surface area, which represents less than half that observed in the *Drosophila* complex[24,25]. *Drosophila* Spz possesses a five-residue

insertion (Asn-59-Gln-62) in the vicinity of mosquito Cys-59, which expands its interface (Supplementary Fig. 3). Nonetheless, Spz1C adopts an asymmetric binding mode, reminiscent of that found in *Drosophila* despite the lack of sequence conservation (Supplementary Fig. 4), in which chain P contributes about 610 Å² and chain D, 280 Å² to the concave binding site. The interactions of Spz1C chain P at the concave site involves mostly residues from the first and third β-strand of Spz1C (Gln-14 and Leu-16, and Tyr-71, respectively) with the LRRNT1 of Toll5A (Tyr-79[A], Tyr-65[A] and His-54[A], respectively). Hydrogen bonds occur between His-18 in the Spz1C Trp-loop and the Phe-60 main-chain carboxy group in the receptor. Spz1C Tyr-38[P] and Toll5A Tyr-56[A] interact with side chain-mediated contacts (Fig. 2). There is a salt bridge between Arg-156[A] in Toll5A LRR2 and Glu-11[P] in Spz1C. The same residue in chain D is solvent-exposed and not well resolved in density. Glycans linked to Asn-521[B] on LRR17 restrict Spz1C spatially in the vicinity of Arg-50 and Gln-51 in chain D. As most side chains are poorly resolved, we generated a full-atom complex upon side chain modelling, which is suitable for surface analysis. Neither the electrostatic and hydrophobic networks of interactions (Supplementary Fig. 5) nor the position of interacting glycans (Supplementary Fig. 3) are conserved between mosquito and *Drosophila* complexes.

The overall structure of Spz1C is however, very similar to DmSpz, if the flexible Trp-loops between residues His-18 and Gln-40, and the β-wings between residues Tyr-71 and Val-89, are omitted. Indeed, the R.M.S.D of superimposed Cys-knot domains is about 2.2 Å over 53 C α atoms in the absence of these flexible regions[6,24,25,34]. Interestingly, in the mosquito Toll5A-Spz1C complex, these loops define asymmetric contacts at the dimerization interface with the "back" of chain B (Fig. 2). Chain D of Spz1C binds extensively to the dimerization interface with over 1,300 Å² of its accessible surface area, while chain

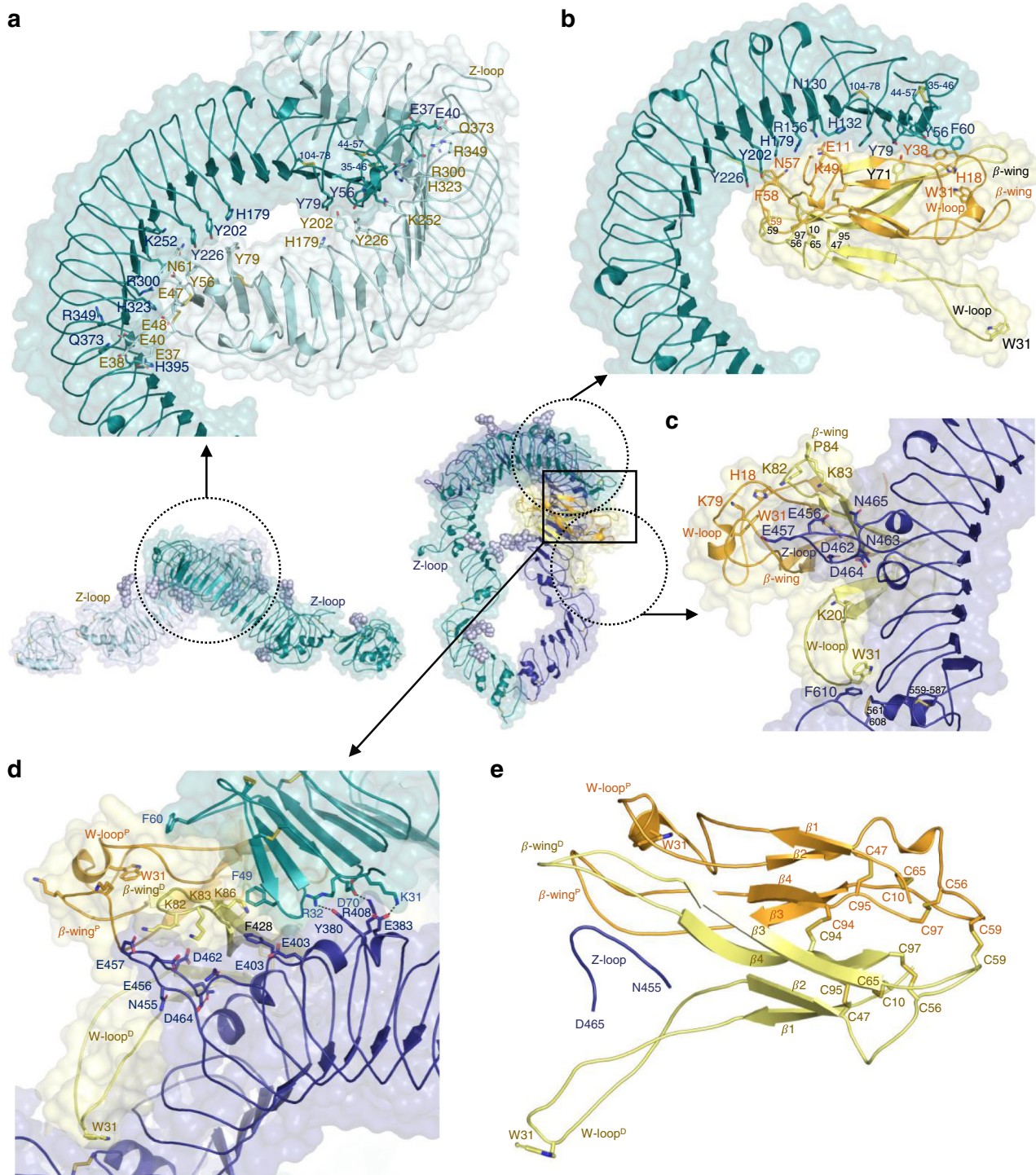

**Fig. 2 | Ligand-induced asymmetry. a** Close-up top view of the homodimer interface with N-terminal receptor-receptor interactions at LRRNT1-LRR14 and accessible Z-loops. **b** Zoomed-in side view Spz1C concave binding to LRRNT1-LRR7. **c** Spz1C convex binding at the dimerization interface. Spz1C mediates extensive contacts between β-wings and the Toll5A Z-loop located between LRR14-LRR15 (Asn455-Asp465), with the distal Trp-loop stabilizing the hinge region between LRR domains. **d** Receptor-receptor contacts persist within the asymmetric hetero-complex. **e** Asymmetry in Spz1C. The proximal Trp-loop adopts a closed helical conformation with a buried Trp-31 that caps the primary receptor LRRNT1, both β-wings and the Z-loop, while the distal Trp-loop is extended and deviates from the core of the molecule, contributing to the asymmetry of the heterodimer complex.

P contributes only 180 Å$^2$, confined to a loop that protrudes from the convex side of LRR14. Remarkably, the integrity of the LRR14 loop seems to determine the dynamic interactions between Toll5A and Spz1C, while its cleavage stabilises the complex (Supplementary Figs. 6 and 7). This property is reminiscent of the Z-loop of nucleic-sensing TLRs[22,35–37]. We will therefore refer to it as the Z-loop with conserved Asn and Asp residues suitable for Asparagine endopeptidase (AEP) processing[38,39], while noting the absence of a cathepsin site in Toll5A.

In the crystal structure of refolded DmSpz, the β-wings (residues 75–93) are symmetric in both protomers, displaced from the Cys-knot framework by 90° and participating in crystal packing[34]. By contrast in our cryo-EM structure, the β-wings adopt a very different conformation contacting the Z-loop in a pincer-like grasp (Fig. 2c–e and

Supplementary Fig. 6A). Molecular surface analysis of the receptor and the ligand suggests that the electrostatic charge distribution and shape complementarity are suitable for receptor-ligand coupling upon conformational selection (Supplementary Fig. 5). The dimerization interface of 1480 $Å^2$ is one and a half times the area of the concave binding surface, suggesting a higher affinity binding compared to the concave interface, while the absence of particles with a stoichiometry of 1:1 may indicate a sequential mechanism of assembly in nonequilibrium unsaturated complexes.

Interestingly, Spz1C possesses highly asymmetric Trp-loops. The distal one is extended and participates in dimerization with Trp-31$^D$ stacking against Phe-610$^B$ at the junction between LRR domains. The other loop adopts a short helical structure, which buries Trp-31$^P$ in a hydrophobic network that includes receptor N-terminal residue Phe-60$^A$ (Fig. 2). It is possible to model Spz1C with both protomers in a closed conformation based on a proximal template. Such a conformation results in a perfectly symmetric ligand able to bind Toll5A but unable to dimerise due to clashes between $\beta$-wings and the Z-loop. Equally, it is possible to model a fully extended Spz1C based on the distal protomer as a template. The resulting structure presents $\beta$-wings open wider for Z-loop interactions than the asymmetric ligand. More importantly, it exposes Trp-31$^P$ for potential interactions with the N-terminal cap of the primary receptor A, which might be required for fully dissociating asymmetric receptor-receptor contacts remaining in this area.

The N-terminal region of Toll5A (chain B) deviates by an angle of ~60°, compared to the axis traversing chain C from N- to C-terminus (Fig. 1). There is evidence of extra density in the original 2:1 heterodimer map that matches the dimensions of a Cys-knot domain without its loops. This indicates that Spz loops and wings only adopt a stable conformation upon receptor dimerization. As there is no possible contact at the dimer interface for the second ligand, we propose that the asymmetric disposition of the receptor chains is the molecular basis of negative cooperativity in the system, where binding of the first ligand in non-saturating conditions makes binding of a second molecule energetically unfavourable. In the presence of saturating ligand concentrations, symmetric ligand binding may occur, in which both ligands can make reciprocal interactions with the receptor heterodimer.

## Toll5A ectodomain competes with ligand binding

A third particle was observed with a stoichiometry of 3 receptors and 1 ligand (Supplementary Figs. 1 and 2). It is a hybrid of the homodimer (Toll5A chains A and C) and the 2:1 Toll5A-Spz1C heterodimer (Toll5A chains A and B) leading to a trimer of A, B and C chains, with Spz1C bound to the back of chain B and the concave side of chain A without any further contacts with chain C. Some minor differences are observed in main-chain and side-chains positions, which might be due to flexibility in both receptor and ligand molecules and the intermediate resolution of the heterotrimer map at 3.7–4.7 Å, compared to the homo- and heterodimer. While binding of a first Spz1C ligand shifts the receptor to adopt an asymmetric dimer configuration, the second binding site remains predominantly associated to another receptor chain.

## Receptor specificity is achieved with low ligand binding affinity

The interaction between Spz1C, and the ectodomain of Toll5A was characterised using a range of biophysical techniques. We used surface-plasmon resonance to measure the kinetics of association and dissociation between mosquito and *Drosophila* Toll and Spz proteins (Fig. 3A and Supplementary Fig. 8) and found that Toll5A only binds Spz1C. Surprisingly, *A. aegypti* Spz1A did not bind Toll1A, despite being the closest structural homologues of DmToll1 and DmSpz1 in the mosquito. Spz1A did not bind Toll5A, and neither did DmSpz1. Furthermore, Spz1C did not bind Toll1A or DmToll1, suggesting

receptor-ligand specificity. By contrast, Toll5A and Spz1C interact with a $K_D$ value of ~2 μM, whereas *Drosophila* Toll binds Spz with a much higher affinity of 30–80 nM, consistent with previous studies[40]. Hence, the Toll5A-Spz1C complex is species- and paralogue specific despite being of low affinity.

## Ligand binding triggers a conformational change within Toll5A dimers in solution

We then used SEC-MALS experiments in the presence and absence of Spz1C to characterise the stoichiometry of the receptor and its complex. This technique revealed a concentration-dependent shift in the stoichiometry of the receptor, from a monomer to a homodimer in the absence of ligand when the concentration was increased from 20 to 50 μM prior to size-exclusion chromatography (Fig. 3). In the presence of 20 μM Toll5A saturated with Spz1C, a complex formed that was polydisperse but at 50 μM Toll5A saturated with Spz1C, the mixture appeared monodisperse with a mass corresponding to 206 kDa. SEC-MALS indicates a mass of 175 kDa for the unbound receptor dimer so the 206 kDa form is consistent with a 2:1 complex of Toll5A and a disulfide-linked Spz1C dimer (26 kDa).

We then explored this 2:1 complex further using SEC-SAXS at 50 μM and compared it to the receptor homodimer at the same concentration and in the same buffer. SAXS allows rapid assessment of structural changes in response to ligand binding and can quantitatively characterize flexible molecules. The Guinier plot demonstrates the aggregation-free state of the receptor and its complex (Supplementary Fig. 9). As one of the few structural techniques amenable to dynamical systems, SAXS analysis suggests that the 2:1 complex is less flexible than the homodimer, despite its slightly larger dimensions (Supplementary Table 2). Hence, the receptor likely undergoes conformational changes upon ligand binding.

## Toll5A-Spz1C complexes undergo dynamic exchange

The linearity of the Guinier plot (Supplementary Fig. 9) does not ensure the ideality of the sample, hence further direct methods were used in solution and under native conditions. We used Analytical Ultracentrifugation (AUC) to determine relative concentrations, sedimentation coefficients, molecular weight and shape (frictional ratio) of Toll5A and Toll5A-Spz1C complexes (Supplementary Fig. 10). These experiments reveal that Toll5A ectodomain alone is in equilibrium between monomers (≤ 3 μM) and dimers that prevail at concentrations ≥ 30 μM (Fig. 4A). The concentration-dependent stoichiometry differs from SEC-MALS, for which monomeric receptor was observed at 20 μM, most likely as a consequence of sample dilution or the effect of the matrix on protein-protein interactions during the size-exclusion chromatography step that precedes MALS.

The shape of the AUC curves indicates that receptor-receptor interactions undergo slow exchanges, defined by discrete peaks at 5.6S, and 7.3S, respectively. In contrast, the presence of Spz1C at equimolar concentration causes the formation of Toll5A-Spz1C complexes with sedimentation coefficients ranging between 5.5 and 8.6 S (Fig. 4b). Toll5A forms heterogenous complexes with its ligand, which may include a 1:1 complex at 6.7S, as well as 7.3S (homodimers) and 7.8S species (2:1 heterodimers) in different conformational states. However, in the presence of excess Spz1C, better resolved molecular species sediment as two distinct populations of 6.7S and 8.5S. These complexes are reminiscent of the 1:1 and 2:2 complexes formed by DmToll and Spz in similar experimental conditions[23,25].

To determine whether these conformers are present for mosquito proteins at more physiological concentrations of ligand, we measured the composition of Toll5A-Spz1C mixtures at nanomolar concentrations using mass photometry (Fig. 4c, d). Our experimental setting was able to measure accurately molecules and complexes above 60 kDa, while free Spz1C was below the threshold, which resulted in erroneous mass determination. At 25 nM Toll5A in the presence of 25 nM Spz1C,

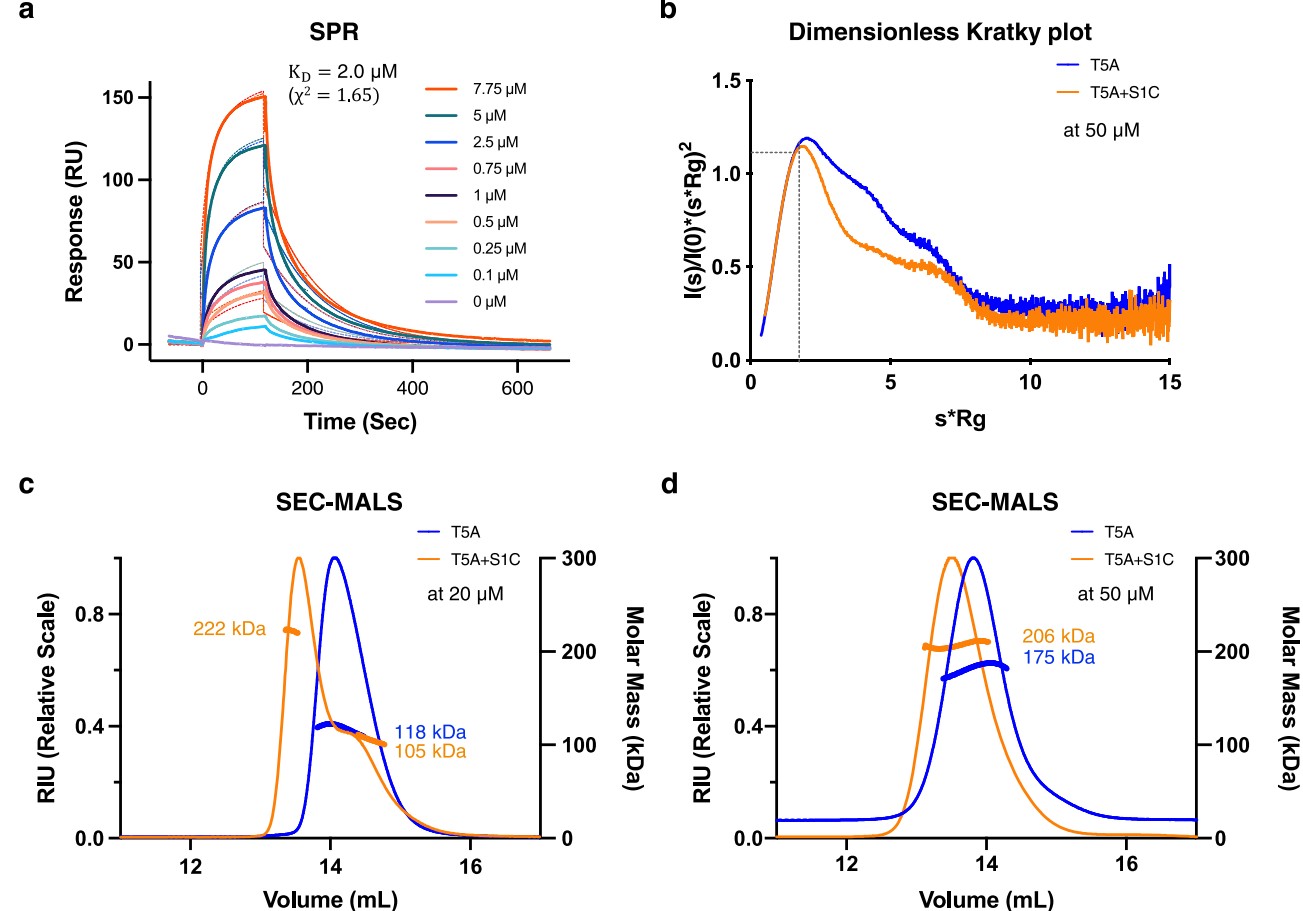

**Fig. 3 | Spz1C low affinity binding to Toll5A homodimer decreases protein complex flexibility. a** SPR binding analysis. Sensorgrams of Toll5A (acronym, T5A) run over a range of concentrations (0.1–7.5 μM) on an amine coupled Spz1C-chip (acronym, S1C). **b** SEC-SAXS dimensionless Kratky plot analysis at 50 μM. SEC-MALS analysis at 20 μM (**c**) and 50 μM (**d**) of Toll5A on its own (blue) and in complex with Spz1C (orange). Both MALS and SAXS were carried out upon loading ~50 μl samples at the given concentrations onto a Superose 6 size-exclusion chromatography column (GE Healthcare) in 50 mM Tris pH 7.5, 50 mM NaCl.

Toll5A is predominantly monomeric but a 2:1 complex is detected along with a species at 48 kDa, corresponding most likely to Spz1C. By contrast, at 50 nM Toll5 in the presence of 50 nM Spz1C, 1:1 ligated monomer and 2:2 heterodimer complexes appear in addition to Toll5A monomer and 2:1 complex. Hence, mass photometry detects stoichiometric conversions in non-equilibrium conditions. Toll5A homodimer is not detected consistent with the AUC data that suggests a $K_D$ value for this interaction in the μM range. None of these techniques detected the heterotrimer, which is therefore most likely a side effect of cryo-EM's capacity of visualising transition intermediates.

## Spz1C drives the production of antimicrobial peptides

In order to assay the activity of Spz1C, we stimulated the *Aedes aegypti* cell line Aag2, which constitutively expresses Toll5A (Supplementary Fig. 11), with either full-length Spz1C proprotein or processed forms and Gram-negative bacteria for IMD pathway stimulation, as a positive control for a potent innate immune response. RT-qPCR was used to measure the induction of a range of antimicrobial peptides.

Aag2 cells do not up-regulate antimicrobial peptides (AMP) upon stimulation with pro-Spz1C. In contrast, processed Spz1C potently stimulated the production of several antimicrobial peptides, including Defensin 1, Cecropin A, Glycine-rich repeat protein (GRRP) holotricin and Attacin B, (Fig. 5A). Using RT-PCR, we found all members of the Spz family to be constitutively expressed in Aag2 cells, except Spz2 (Supplementary Fig. 11). Given that Aag2 cells were shown to be unresponsive to microbial stimulation of the Toll pathway[41,42], our result confirms full Toll signalling capacity when provided with the active

ligand. Aag2 cells may therefore be defective either in microbial sensing or within the protease cascade activating Spz.

Upon activation, we found that Spz1C signalling overlaps but is distinct to that induced by a soluble extract of heat inactivated Gram-negative bacteria (GNB), which in addition activates the production of Gambicin, a general-purpose AMP. Neither GNB or Spz1C triggers the production of Vago, which is regulated by the RNA interference pathway, or Diptericin, an AMP potently induced by GNB in *Drosophila* via the IMD pathway[43].

Next, we assayed production of GRRP holotricin, an AMP that is strongly induced by Spz1C, in a dose-response experiment (Fig. 5b). Activation of holotricin expression occurs over a wide range of concentrations with an $EC_{50}$ at sub-nanomolar concentrations. Furthermore, Spz1C signalling displays the same hallmarks of negative cooperativity observed in the *Drosophila* pathway, with 10 % to 90 % maximal signalling requiring an increase in ligand concentration of about 600-fold.

Interestingly, we found that different Spz paralogues have different signalling activities, as suggested by the capacity of *Aedes aegypti* Spz5 to potently activate gambicin (GAM) (Fig. 5c). While *Drosophila* Spz1 is able to stimulate GAM moderately in Aag2 cells, this property is not shared by Spz1C. In *Drosophila*, Spz5 is recognised by Toll1, Toll6 and Toll7[9–11]. Given that Aag2 cells express other members relating to the Toll family of receptors (Supplementary Fig. 11), it is conceivable that one of them triggers GAM activation reflecting its ligand specificity as opposed to the documented promiscuity of the *Drosophila* system.

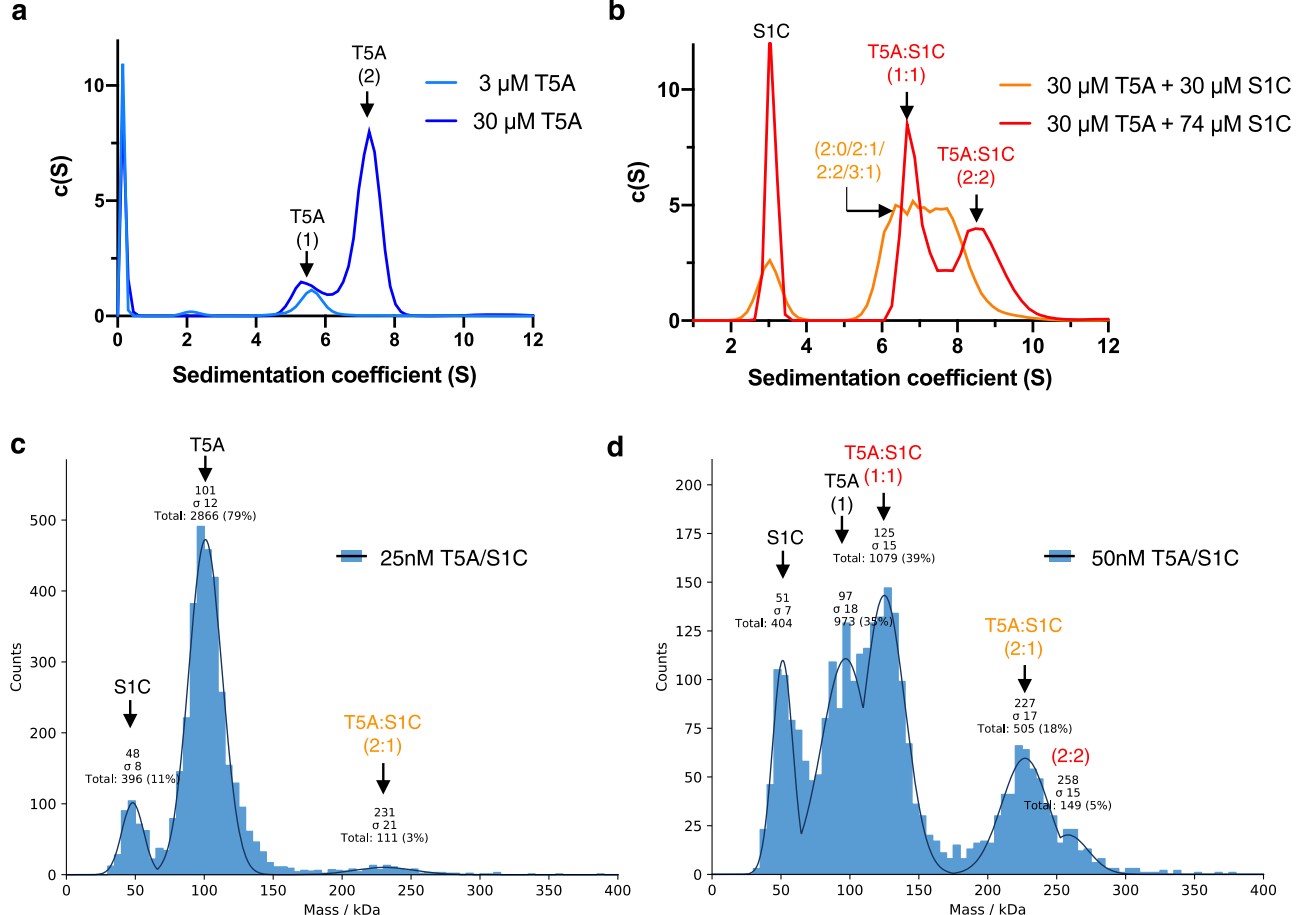

**Fig. 4 | Slow exchanges in receptor-receptor interactions in the absence of ligand contrast with fast dynamics in the presence of ligand.** AUC sedimentation velocity profiles and the shifts in sedimentation coefficients are indicative of dynamical behaviour or the receptor in the presence and the absence of its ligand (**a–b**). Mass photometry (**c–d**) reveals the concentration-dependent stoichiometries of Toll5A and Spz1C. The single-ligated Toll5A dimer is detected at the lowest measurable concentration of 25 nM (**c**), whereas saturated dimer and ligated monomer appear at 50 nM and above (**d**).

## Spz1C regulates genes involved in immunity and homeostasis

We have used RNASeq to define the transcriptomic signature linked to Spz1C and compare it to activation of the immune system by Gram-negative bacteria and purified DAP-PGN, which in *Drosophila* activates the IMD signalling pathway. As shown in Fig. 6 there are 85 genes regulated by Spz1C, of which 83 are up-regulated and only 2 are down-regulated. These include GRRP holotricin, confirming its role as a marker of Spz1C, as well as other molecules linked to complement and immunity. Many genes encoding Clip Serine proteases, Serpins and Gram-negative Binding Protein (GNBP) are upregulated. In *Drosophila* these proteins function in peptidoglycan recognition and proteolytic cascades that activate Spz.

The Patched 1 (Ptc1) 7-TM receptor is strongly induced with potential implications for tissue regeneration[44]. The ligand for Ptc1 is Hedgehog, a morphogen involved in embryonic segmentation. This signalling pathway also plays a crucial role in maintaining adult tissue homeostasis. It is not known whether Ptc1 functions in insect immunity. Another upregulated gene encodes a putative ecdysone inducible protein ortholog L2 and is homologous to *Drosophila* Imaginal morphogenesis protein-Late 2 (IMPL2)[45], which is involved in the regulation of metabolism, growth, reproduction and lifespan. Two genes are down-regulated: Fatty Acyl-CoA Reductase (FAR) and the prostaglandin EP4 receptor. Interestingly, eicosanoid biosynthesis is controlled by the Toll pathway in some insects and also human PGE2 triggers a negative feedback loop, in which TLR4 signalling is restricted.

## Discussion

In this study we show that the haematophagous mosquito *A. aegypti* has a duplicated Toll ligand, Spz1C, apparently unique to the *Aedes* genus (Supplementary bioinfomatic analysis and Supplementary Fig. 12), that specifically activates the Toll5A paralogue with low affinity. We provide direct evidence of their interaction using purified proteins and show that ligand binding may be restricted by receptor self-association in a dose-dependent manner, resulting in conformational intermediates that we visualise by single particle cryo-EM. Functionally, Spz1C and Toll5A are known sensors of fungal infections in the fat body of adult mosquitoes[26]. Interestingly, anti-dengue activity has been assigned to the entomopathogenic fungus *Beauveria bassiana*[29], and to the Toll pathway[27,28], without elucidating the potential roles of individual Toll receptors or their respective ligands.

Here we provide evidence of a robust Spz1C response in Aag2 cells that differs from its *Drosophila* orthologue and mosquito paralogue. We also note that duplicated Toll and Spz are more divergent in sequence than 1-to-1 orthologues, with Spz1 the only duplicated cytokine within its family, while duplication occurred for members of the Toll family including Toll1, Toll5 and Toll9 and expansion of the Toll6/7/8 family with two additional members, Toll10 and Toll11 with potential anti-plasmodial properties[46]. *Aedes aegypti* diverged from *Anopheles gambiae*, which is the main vector for malaria, about 217 Myr ago. *Aedes aegypti* efficiently vectors flaviviruses such as Dengue, as evident from the large number of nonretroviral integrated

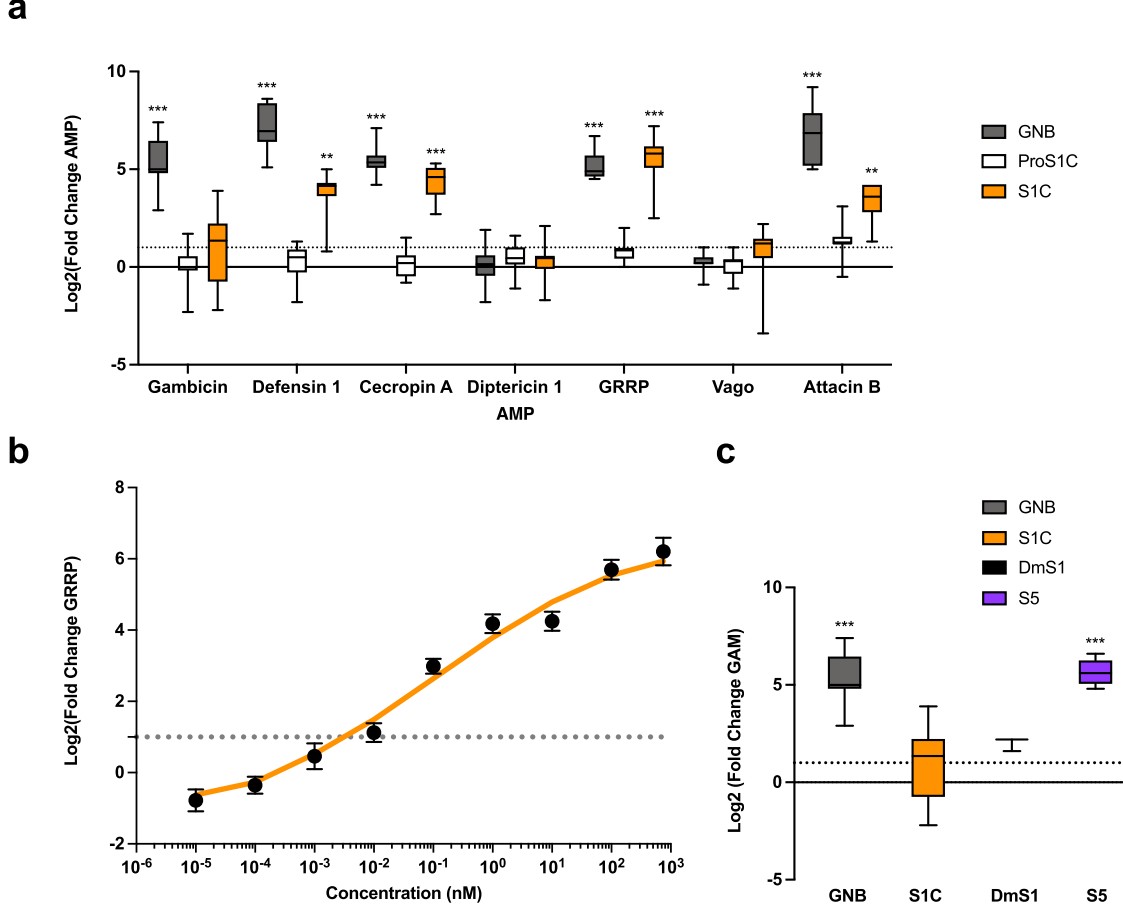

**Fig. 5 | Cleaved but not proprotein Spaetzle1C activates antimicrobial peptide production.** Spz1C activates a set of antimicrobial peptides (**a**), over a large range of concentrations (**b**), and in an isoform-specific manner (**c**). Expression of antimicrobial peptides in Aag2 cells at 12 hours after stimulation with heat-inactivated extract from Gram-negative bacteria (abbreviated GNB) or Spz1C (proS1C and S1C: before and after cleavage of the pro-domain, respectively) upon addition of purified protein at 100 nM in the media and RT-qPCR. Data are presented as box plot with whiskers min to max values. ($n = 12$ biologically independent experiments–Except "Vago" condition where $n = 9$) (**b**) Kinetics curves of GRRP expression in Aag2 cells after stimulation with different concentration of Spz1C (S1C). The fitted curve (in orange) represents smoothed conditional means calculated by ggPlot2 on R. Data are mean ± SEM. ($n = 18$ to 36 biologically independent experiments for each concentration). **c** Differences in Gambicin (Gam) expression upon Spz1C, *Drosophila melanogaster* Spz1 (DmS1), and *Aedes aegypti* Spz5 (S5)

stimulation. Heat-inactivated *E.coli* extract (GNB) was used as a positive control and conditioning buffer was used as a negative control. Data are presented as box plot with whiskers min to max values. (n = 12 biologically independent experiments for "GNB" and "S1C" condition, $n = 9$ for "S5" condition and $n = 3$ for "DmS1" conditions). For all experiments, expression values (Log2(FC)) were determined using the ΔΔCT method with normalisation to mRNA levels of the eEFG1a housekeeping gene. The dotted line indicates a value of Fold Change=2 as we determined to the limit to have change in expression profile. All Statistical analyses were performed in R using one-sided Student T-test or Kruskal–Wallis test (according to application condition of each after testing normality and equality of variances of each condition) to compare result versus a FC > 2 with significance $p$ value defined > 0.05. Stars indicate significance: * = $p$ value < 0.05, ** = $p$ value < 0.01, and *** = $p$ value < 0.001. Source data are provided as a Source Data file.

RNA viruses present in the *A. aegypti* but not the *A gambiae* genome. Interestingly, the Asian tiger mosquito *A. albopictus*, which diverged from *A. aegypti* more recently, further expanded *Spz1C* genes. In contrast, more divergent species from the *Anopheles* and *Culex* genera, which are not (or less) competent to vector these viruses, do not possess this paralog. However, a direct role of *Spz1C* in vectorial competence enabling flaviviruses, such as DENV, to exist symbiotically at high titre in mosquitoes remains to be established.

Our biophysical characterisation shows that Spz1C binds specifically to Toll5A with micromolar affinity compared to the nanomolar binding of DmSpz1 by DmToll1. DmSpz1 is also able to bind promiscuously to Toll6 and Toll7 and so it may be that low affinity binding confers signalling specificity for Toll5A. Consistent with this idea the *Aedes* Spz5 paralog but not Spz1C activates the AMP gambicin in Aag2 cells as strongly as bacterial extracts while DmSpz1 induces a partial activation (Fig. 5c). Thus *A. aegypti* has evolved a tiered Toll mediated immune system compared to *Drosophila* where Toll1 alone fulfils most immune functions.

Low affinity ligand binding and its ramifications in terms of signalling has been extensively characterised for mammalian cytokine receptors for interferons and interleukin ligands[47–49]. Such cytokines display pleiotropic effects while inducing a spectrum of redundant and yet distinct cellular functions. Receptor-ligand association and dissociation rates ($k_{on}$ and $k_{off}$) have been found to be key in determining signalling outcomes. On-rates determine the amount of STAT transcription factor activation upon controlling the number of ligand-receptor complexes formed at the plasma membrane. In contrast, off-rates correlate to the kinetics of STAT activation depending on the half-life of ligand-receptor complexes. Alternatively, cell surface abundance of cytokine receptors plays a major role in triggering different transcriptional programs and cell fates, through obeying the mass action law and titrating cytokine concentrations. Our hypothesis in the case of *Aedes* Toll5A and Spz1C is that cell fate will be regimented in two ways depending on receptor and ligand concentrations: (i) ligand concentration changes the intensity of the signal (increased Spz1C for increased antimicrobial peptide production as illustrated in

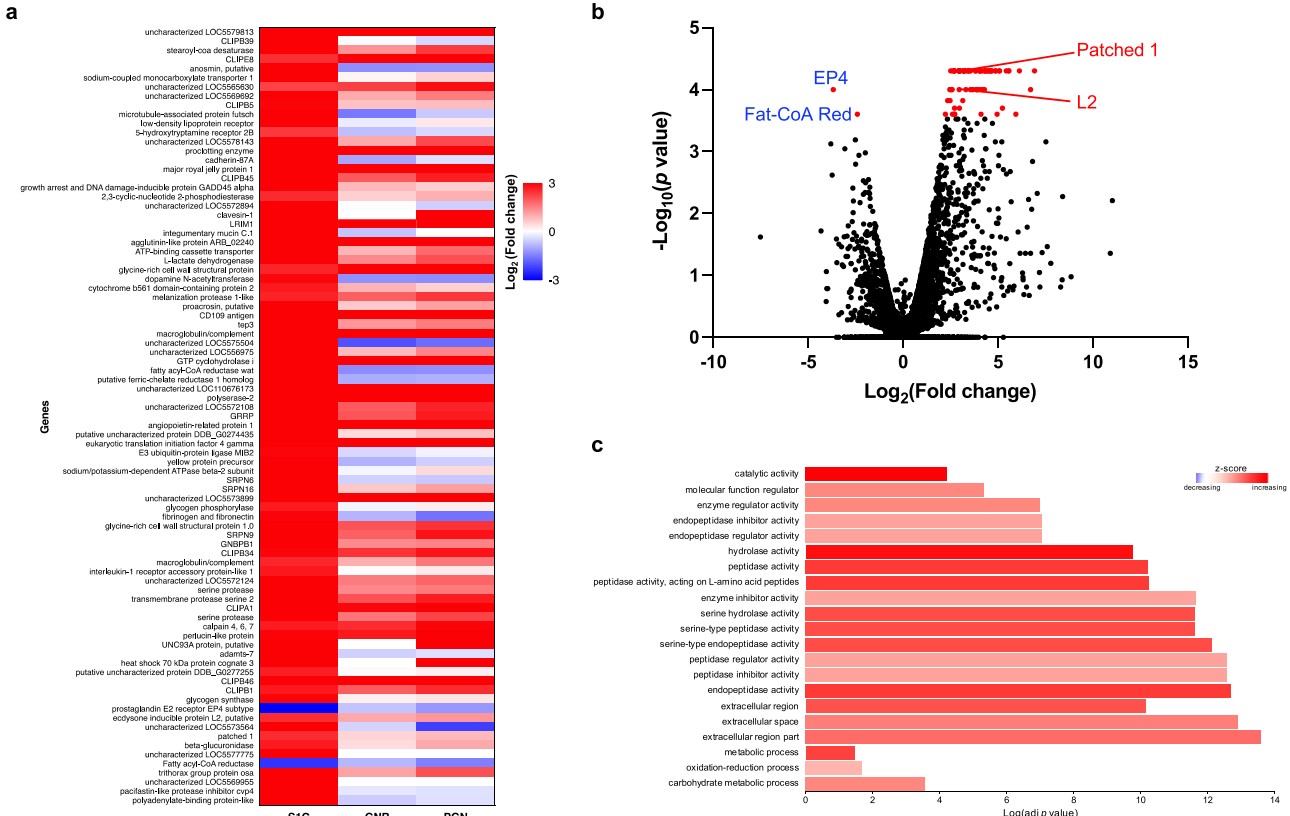

**Fig. 6 | Spz1C transcriptional signature in Aag2 cells by RNAseq analysis. a** Heat map of genes exhibiting different responses to S1C stimulation compared to MOCK-stimulated Aag2 cells. Corresponding gene expression is also shown for Gram-negative bacteria (GNB) and PGN stimulation. **b** Volcano plot of individual two-sided pairwise comparisons of S1C versus MOCK-stimulated Aag2 cells expression. Significantly differentially expressed genes (DE-genes) are in red, non-significant genes are in black. Data in **a** and **b** are represented using CummeRbund

package 2.8.2 for visualization of CuffData objects generated by Cufflinks package 2.2.1. **c** Gene ontology (GO) term enrichment analysis. The GO annotation results were based on 84 genes identified by RNAseq (**a**). Gene ontology categories included molecular function (MF), cellular component (CC) and biological process (BP). GO categories for each function are sorted by increasing order of evidence, based on the GO enrichment test *P* value (<0.05) used by PANTHER pipeline 16.0.

Fig. 5); and (ii) increased receptor density might promote inhibitory self-association leading to immune quiescence.

The structures we present provide a plausible molecular explanation for negatively cooperative signalling. Binding of the first Spz1C homodimer to form the asymmetric complex can occur transiently at relatively low ligand concentrations but the 2:2 form would require saturation, and hence only form at saturating ligand concentrations. There are two classical theoretical models of sequential binding that lead to negatively cooperative receptor signalling as initially proposed by Koshland[50,51]. Binding of the first monomeric ligand to a dimeric receptor partially activates signalling but full activation requires the less favourable binding of the second ligand monomer. In the second model, the binding of two monomeric ligands to form a 2:2 complex is required for signal transduction. The two models lead to somewhat different theoretical stimulus-response curves and our experiments appear to fit better with Model 1 for the production of antimicrobial peptides (Fig. 5). If the ligand is limiting there is an ultrasensitive response with a pronounced threshold, more reminiscent of positive than negative cooperativity[52]. In our cell culture assays, Spz1C is likely not limiting, however this phenomenon may be relevant for development and homeostasis controlled by Spz in the whole organism.

The activation profile of Spz1C differs from *Drosophila* Spz1 and *Aedes* Spz5, despite their shared cystine-knot fold, having a transcriptomic profile that indicates functions that go beyond immunity. Spz1C specifically activates the production of Hedgehog receptor, insect insulin-binding protein L2 and genes involved in reprogramming fatty acid metabolism. If Spz1C and Toll5A were both upregulated

in the salivary glands, and achieved the double ligated state as suggested by Bonizzoni et al.[53], it is also conceivable that Spz1C and its receptor could be involved in viral tolerance by promoting lipid metabolism and homeostasis. The mechanism of action may differ in the midgut where Spz1C expression is induced to limit viral invasion, an immunological role[31]. By contrast in the salivary gland[30], where Toll5A up-regulation may increase avidity for the ligand, a double ligated and hence fully saturated receptor complex might lead to a signal that differs from low density single-ligated receptors. Hence, future studies need to tackle the signalling capacities of 2:1 and 2:2 complexes by detecting the recruitment of MyD88 first, and if so, by checking if different sets of genes are controlled by both type of complexes. Fluorescence microscopy experiments will be best suited to detect such complexes along the presence of unliganded receptor homodimers to distinguish between these models.

Remarkably and despite a significantly lower affinity, Spz1C is as potent in stimulating mosquito Aag2 cells as DmSpz is at stimulating *Drosophila* S2 cells. An alternative explanation is that cleavage of the Z-loop promotes stable ligand binding and thus processing might account for the discrepancy between protein affinity and cellular potency of Spz1C. On the other hand, Spz1C binding prevents cleavage of Toll5A's Z-loop, a finding that is not consistent with the Z-loop impeding ligand binding. Additionally, we expressed Toll5A in insect cells without detecting such processing, which likely rules out its spontaneous occurrence. However, *Drosophila* Toll1 can be cleaved at an equivalent residue Asp 458, and forms a stable dimer with the ectodomain remaining intact[23]. Taken together, we conclude that this

region is critical for promoting ligand-induced dimerization while the physiological relevance of its processing requires further examination. Of note, the nucleic-acid binding TLRs that are activated by endoproteolytic cleavage display positive cooperativity[54].

In light of the above it is likely that there will be a degree of synergy between Toll5A signalling and the vectorial capacity of *Aedes aegypti*. Our study sheds new light on Toll signalling, while raising fundamental questions. Do Toll receptors undergo endoproteolytic processing to regulate their activity? Which oligomeric forms occur in vivo and what are their respective signalling outputs? More importantly, can Spz1C signalling be exploited to fight mosquito-borne diseases? If so, the structural data in hand can guide future transmission-blocking strategies.

## Methods

### Bioinformatic analysis

Mosquito Toll and Spz sequences were retrieved via BLAST searches in Uniprot[55] and Vectorbase[56]. Sequence alignments were performed using Muscle[57]. Homology modelling was carried out using Modeller version 9[58]. Alignments were visualised in Jalview[59] and 3D-models in PyMol (Molecular Graphics System) and Chimera[60].

### DNA constructs

*Aedes aegypti* genomic DNA from 5 male and 5 female mosquitoes was a kind gift from Dr Emilie Pondeville (University of Glasgow, UK). Constructs were either derived from genomic or synthetic DNA upon codon optimisation to improve the protein production yields for Toll1A (Vectorbase identifier AAEL026297), Toll5A (AAEL007619), Spz1A (UniProt accession code Q17P53), Spz1C (UniProt accession code Q16J57; Vectorbase identifier AAEL013433) and SpzX (AAEL013434). Constructs were cloned into baculovirus transfer vector pFast-Bac1 within BamHI and NotI, and into pMT-V5-His$_A$ within KpnI and NotI (ThermoFisher).

### Cell culture

Aag2 cells were a kind gift from Prof Alain Kohl (University of Glasgow, UK), and were maintained in Schneider's Insect Medium with L-glutamine and sodium bicarbonate (Merck KGaA, Darmstadt, Germany) supplemented with 10 % (v/v) foetal bovine serum (FBS - Sigma-Aldrich, USA), 100 U/ml penicillin and 100 µg/ml streptomycin at 28 °C in a humidified atmosphere without $CO_2$.

Sf9 cells (ThermoFisher) used for baculovirus preparation were maintained in Insect- XPRESS™ Protein-free Insect Cell Medium with L-Glutamine (Lonza) at 28 °C under agitation. S2 cells used for stable insect expression were a gift from Prof Jean-Luc Imler (University of Strasbourg, France). They were maintained in the same medium at 28 °C with or without agitation.

### Protein production and purification

**Spz Proprotein and Cys-knot domain preparation.** *Aedes aegypti* Spz paralogues were produced in a baculovirus expression system (Bac-to-Bac, ThermoFisher) with a C-terminal Strep-tag® II with or without an engineered TEV-cleavage site between the pro-domain (Spz1C pro-domain residues 43–218) and the Cystine-knot domain (Spz1C residues 219–320) after establishing the domain boundaries using limited trypsin proteolysis.

Typically, 4 litres Sf9 at 2 million cells per ml were infected at a MOI = 2.0 and cultured under agitation at 19 °C instead of 27 °C for optimal expression over 5 days. The supernatant was harvested after removing cells by centrifugation at 2000 g for 10 min and filtered on a Sartobran P sterile capsule of 0.45 µm (Sartorius). The buffer was then exchanged to buffer A (150 mM NaCl, 100 mM Tris-HCl, pH 7.5, 1 mM EDTA) and concentrated to 500 ml using a Centramate tangential flow filtration system (Pall). It was loaded on a Strep-Tactin®XT Superflow® resin (IBA) following manufacturer recommendations. The resin was

equilibrated in 10 column volumes of buffer A prior to use and subsequently washed with 10 column volumes of the same buffer to remove non-specifically bound proteins. Strep-tagged protein was eluted in buffer A containing 50 mM biotin. Peak fractions were pooled and purified by anion-exchange on a 5 ml Hitrap Q (Cytiva, formerly GE Healthcare Life science) in a NaCl gradient from 50 mM to 1 M, 50 mM Tris-HCl, pH 7.5, followed by size exclusion chromatography on a Superdex 75 10–300 GL column (Cytiva) at 0.5 ml/min in buffer C: 50 mM NaCl, 50 mM Tris-HCl, pH 7.5. Fractions were analysed by Coomassie-stained SDS-PAGE. Protein concentrations were quantified by absorption at 280 nm. A typical yield was 3 mg of purified protein per litre of cell culture. *Drosophila* Spz production has been described elsewhere[61].

**Toll ectodomain preparation.** A stable Schneider 2 cell line containing pMT-V5/His-A *Aedes aegypti* Toll1A (AAEL026297) ectodomain (residues 1–835) and Toll5A (AAEL007619) ectodomain (residues 1- 789), a C-terminal TEV-cleavable Protein A fusion and a Flag-tag was induced with copper sulfate 0.5 mM at 3 million cells/ml. Cells were cultured at 27 °C under agitation and harvested after 3–4 days. Culture supernatant were filtered, buffer-exchanged and concentrated to 0.5 L in 150 mM NaCl, 50 mM Tris-HCl pH 7.5, 0.05% Tween 20. Protein A Flag tagged proteins were isolated with IgG Sepharose® 6 Fast Flow (Cytiva), incubated with TEV protease at a 1:10 (w/w) ratio at 4°C overnight in 150 mM NaCl, 50 mM Tris-HCl, pH 7.5 (buffer D). TEV-cleaved proteins lacking Protein A-Flag tag were collected in the flow through upon washing in buffer D. Protein A eluted in 0.1 M Na Acetate pH 3.4. Fractions of interest were pooled and further purified by anion-exchange (as above) and size exclusion chromatography on a Superdex 200 10–300 GL column (Cytiva) at 0.5 ml/min in buffer C.

### Surface plasmon resonance

SPR experiments were performed on a Biacore T200 instrument with dextran-based Sensor Chip CM5 (Cytiva) in 100 mM NaCl, 20 mM HEPES pH 7 running buffer and a flowrate of 30 µl/min. The chips were activated by 1-Ethyl-3-(3-dimethylaminopropyl)-carbodiimide hydrochloride and *N*-hydroxysuccinimide, and Spz1A, Spz1C and DmSpz proproteins and cys-knot ligands were immobilized by amine coupling at pH 4.5. Sensorgrams were recorded and corrected by subtraction of control signal from an empty flow cell. Purified ectodomain analytes of Toll1A, Toll5A and DmToll were injected at concentrations between 0.1 and 7.5 µM in ten-fold dilution series. Kinetic analysis was performed by fitting sensorgrams to a two-state reaction model.

### Analytical ultracentrifugation

Analytical ultracentrifugation experiments were performed on an Optima XL-A/I (Beckman Coulter) centrifuge equipped with a four-hole titanium rotor, double-sector centrepieces, and an interference optical system for data acquisition. Sedimentation velocity runs were performed at 50,000 rpm with 3-min intervals between scans for a total of 190 scans at 20 °C. The sample volume was 400 µL. Data were analysed using Sedfit software version 16.1c[62]. The partial specific volumes, buffer density, and viscosity were estimated using SEDNTERP software (2012-08-23 version)[63].

### SEC-MALS

Size exclusion chromatography-coupled multi-angle light scattering (SEC-MALS) was used to analyse protein monodispersities and molecular weights. SEC was performed using an Äkta Purifier (GE Healthcare) and a Superose 6 10/300 GL column (GE Healthcare) in 50 mM Tris-HCl pH 7.5, 50 mM NaCl. For each measurement, 50 µL of protein at a given concentration was injected and gel filtrated at a flow rate of 0.5 ml/minute. Light scattering was recorded using a miniDAWN TREOS instrument (Wyatt Technology). Protein concentration in each elution peak was determined using differential refractive index (dRI).

The data were analysed using the ASTRA 6.2 software (Wyatt Technology).

## SEC-SAXS

SAXS measurements were performed at Diamond Light Source (Oxfordshire, UK), beamline 21 (B21)[64] at a wavelength 0.89–1.3 Å with a sample to detector distance of 3.7 m and a Eiger 4 M (Dectris) detector, covering a momentum transfer of $0.0026 < q > 0.34$ Å$^{-1}$ [$q = 4\pi \sin\theta/\lambda$, 2 θ is the scattering angle]. The proteins were analysed by size-exclusion chromatography in line with small-angle X-Ray scattering (SEC-SAXS) to avoid the signal from possible aggregates. The samples were applied to a Superose 6 Increase 3.2/300 column (Cytiva) at a concentration of 5 mg/ml and run at a flow rate of 0.075 ml/min in 50 mM Tris-HCl, pH 7.5, 50 mM NaCl. SAXS measurements were performed at 20 °C using an exposure time of 3 s frame$^{-1}$. SAXS data were processed and analysed using the *ATSAS* program package version 2.8.3[65] and ScÅtter version 3.1r (www.bioisis.net). The radius of gyration $R_g$ and forward scattering $I(0)$ were calculated by Guinier approximation. The maximum particle dimension $D_{max}$ and $P(r)$ function were evaluated using the program *GNOM*[66].

## Mass photometry

All mass photometry measurements were executed on a Refeyn OneMP instrument. The calibration was done with a native marker protein standard mix (NativeMark Unstained Protein Standard, Thermo Scientific), which contains proteins ranging from 20 to 1,200 kDa. Coverslips (24 × 50 mm, No. 1.5H, Marienfeld) were cleaned by sequential sonication in Milli-Q water, isopropanol and Milli-Q water, followed by drying with nitrogen. For each acquisition 2 μL of protein solution was applied to 18 μL PBS buffer, pH 7.4 in a gasket (CultureWellTM Reusable Gasket, Grace Bio-Labs) on a coverslip. Increasing working concentrations tested included 25, 50, 75 to 100 nM. Movies were recorded at 999 Hz with an exposure time of 0.95 ms by using the AcquireMP software. All mass photometry movies were processed and analysed in the DiscoverMP version 2.0 software. Samples were measured in duplicates.

## Cryo-EM

Holey carbon grids (Quantifoil Cu R1.2/1.3, 300 mesh) were glow-charged for 60 s at current of 25 mA in PELCO Easiglow (Ted Pella, Inc). Aliquots of 3 μl of between 3–6 mg/ml of Toll5A-Spz1C gel-filtered complex mixed with 8 mM CHAPSO (final concentration, Sigma) were applied to the grids, which were immediately blotted with filter paper once to remove any excess sample, and plunge-frozen in liquid ethane using a FEI Vitrobot Mark IV (Thermo Fisher Scientific Ltd) at 4 °C and 95 % humidity. All cryo-EM data presented were collected eBIC (Harwell, UK) and all data collection parameters are given in Supplementary Table 1. Microscope operations and data acquisition were performed using EPU version 2.1.0.

Cryo-EM images were processed using Warp version 1.0.9[67] and CryoSPARC version 3.3.1[68]. In short, CTF correction, motion correction, and particle picking were performed using Warp. These particles were subjected to two-dimensional (2D) classification in CryoSPARC followed by ab initio reconstruction to generate initial 3D models. Particles corresponding to different classes were selected and optimised through iterative rounds of heterogeneous refinement as implemented in CryoSPARC. The best models were then further refined using homogenous refinement and finally non-uniform refinement in CryoSPARC. Finally, all maps were further improved using ResolveCryoEM in PHENIX version 1.19[69].

The final cryo-EM maps following density modification were used for model building. The crystal structures of *Drosophila* Toll and Spz were used to generate the homology model of Aedes Toll5A and Spz1C using Modeller[58]. The initial model was then rigid-body fitted into the cryo-EM density for the highest resolution map of the apo-dimer of Toll5A in UCSF chimera[60] and manually adjusted and rebuilt in Coot 0.9[70]. Namdinator[71] was used to adjust the structure and several rounds of real space refinement were then performed in PHENIX[72] before the final model was validated using Molprobity[73]. For Toll5A in the holo-dimer and the 3:1 trimer cryo-EM maps, the initial model for the receptor was taken from the apo-dimer refined structure, while Spz1C was the *Drosophila* Spz homology model and the same strategy was applied. All structures were refined and validated before being deposited into the PDB.

## RNA sequencing and RT-qPCR

Total cellular RNA was extracted using the RNeasy Mini Kit (Qiagen, Germany) following manufacturer recommendations. Contaminating DNA was removed using TURBO DNA-free Kit (ThermoFisher Scientific, USA). The quantity and quality of RNA was checked using a NanoDrop spectrophotometer (ThermoFisher) and Bioanalyzer for RNA sequencing samples. Library preparation was performed with TruSeq Stranded mRNA Library Prep Kit (Illumina). Sample sequencing was performed on an Illumina NextSeq 500 using mRNA derived from different conditions (Mock Aag2 cells and stimulated with 100 nM of Spz1C, GNB and PGN). The reads obtained by RNA-seq were analysed using the Cufflinks RNA-Seq workflow against *Aedes aegypti* genome on VectorBase (www.vectorbase.org). All graphic representations were made using CummeRbund package (v3.10) on R. RT-qPCR was conducted in a Rotor-Gene Q system (Qiagen) for over 40 cycles with an annealing temperature of 60 °C, with each well containing 2 μl RNA (10 ng/μl), 0.8 μl 10 μM specific primers (final concentration, 300 nM), 5.4 μl H$_2$O, 1 μl 20X Luna WarmStart RT Enzyme Mix (NEB) and 10 μl 2X Luna Universal One-step Reaction mix (NEB). Samples were measured in triplicates. Assessment of the expression of each target gene was based on relative quantification (RQ) using the comparative critical threshold (CT) value method. The RQ of a specific gene was evaluated in each reaction by normalization to the CT obtained for endogenous control gene *elongation factor 1 alpha* (eEF1a). Three independent infection experiments were conducted. The primers for quantitative RT-PCR used in this study are presented (Supplementary Table 3). Data were analysed by t-test, Mann-Whithney, Krustal Wallis or ANOVA test depending on the application conditions and $P$ value of <0.05 was considered significant on R software.

## Quantification and statistical analysis

Quantification and statistical analyses pertain to the analysis of cryo-EM and RNAseq data are integral parts of algorithms and software used.

## Reporting summary

Further information on research design is available in the Nature Research Reporting Summary linked to this article.

# Data availability

Sequences are available from Vectorbase for Toll1A under the identifier: AAEL026297; Toll5A: AAEL007619; Spz1C: AAEL013433; and SpzX: AAEL013434. Spz5: AAEL001929. Spz1A identifier AAEL000499-PA is obsolete. However, UniProt accession code for Spz1A is Q17P53; Spz1C: Q16J57; SpzX: A0A6I8TFH1; Spz5: Q17JP7; Toll1A: A0A6I8U6W1; Toll5A: A0A6I8TEX2.

The RNA-seq data for this study are publicly available through the European Nucleotide Archive under accession number PRJEB50861. SAXS data were deposited at the Small Angle Scattering database with accession numbers SASDKX8 for Toll5A alone [https://www.sasbdb.org/data/SASDKX8] and, SASDKY8 for Toll5A with Spz1C [https://www.sasbdb.org/data/SASDKY8], respectively. The cryo-EM 3D maps corresponding to the homodimer, the ligated heterodimer and heterotrimer were deposited in EMDB database with accession codes EMD-11984; EMD-11982 and EMD-11983, respectively. The corresponding

atomic models were deposited in PDB with accession codes 7B1D, 7B1B and 7B1C, respectively. There are no restrictions on data availability. Source data are provided with this paper.

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

## Acknowledgements

We thank Drs Katherine Stott, Biophysics Facility manager, and Paul Brear for training in biophysical techniques and continuous support; Drs Marko Hyvönen and Joseph Maman, for helpful discussions. We also thank Joe for training in SEC-MALS and Dr Nathan Cowieson, for supporting the SEC-SAXS at B21 Diamond Light Source. We thank Shilo Dickens, DNA sequencing facility manager, and her team, in particular Dr Markiyan Samborskyy, for RNAseq analysis. We thank Tom Dendooven for sharing the protocol for cryo-EM sample preparation used in this study, and Lee Cooper for Vitrobot training and cryo-EM grid preparation. This work was supported by a New Investigator grant from the Medical Research Council to M.G. (MR/P02260X/1) and Wellcome Trust Investigator Award to N.J.G. (WT100321/z/12/Z).

## Author contributions

M.G. and Y.S. designed biochemical experiments. M.G, S.G.S and Y.S. planned cell assays. M.G., Y.S., and T.H.W. purified and assembled complexes. M.C.K.T. expressed Spz5. M.C.M performed AUC experiments. S.H. and D.Y.C. collected and processed cryo-EM data. M.G. performed model building and structure refinement with the help of M.C.M. M.J.M. carried out MP experiments and analysed SEC-SAXS data. M.G. analysed structural and functional data and wrote the manuscript with N.J.G.

## Competing interests

The authors declare no competing interests.
