## [Peer Review File · Nature Communications]

Structure and dynamics of Toll immunoreceptor activation in the mosquito *Aedes aegypti*REVIEWER COMMENTS

Reviewer #1 (Remarks to the Author):

In insects, Toll receptors are involved in both embryonic development and innate immunity. In *Drosophila*, the only known ligand is the disulphide-linked NGF-like Spaetzle (Spz) protein dimer. In contrast to mammalian Toll-like receptors (TLRs), which are activated via ligand-induced dimerization of their intracellular domains, the precise mechanism of insect Toll receptor activation by Spz remains enigmatic.

The present manuscript therefore presents an important advance to our understanding of this system. The authors present three cryoEM structures of the extracellular domain (ECD) of Toll5A from *Aedes aegypti* in the presence and absence of Spz1C. The 2:1 Toll5A-ECD:Spz1C-dimer complex is of particular interest, as this probably represents an intermediate in the signalling pathway. The structure is supplemented with Aag2 cell-based gene reporter assays.

I found the study very interesting and feel that it is in principle suitable for publication in *Nature Communications*. Having said this, I feel that the structure(s) could/can be better described:

First and foremost, I found the labelling of the chains very confusing, in particular chains B and C in the (probably most interesting) 2:1 complex – better would be B and A respectively (the latter corresponding to Toll in the previous 1:1 complexes, references 28 (Lewis et al., 2013) and 29 (Parthier et al., 2014)). Understanding would be strongly enhanced by renaming “distal” chain E (previously chain L (28) or “trailing” chain J (29)) to e.g. “D” and “proximal” chain F (M / “leading” chain K) as “P”. In addition, the manuscript would benefit from clearer (structural) figures depicting the various interactions. As the biological relevance of the 3:1 complex is questionable, this should be moved to the Supplementary information.

In the 2:1 complex, the N-terminal region of Toll5A chain B (manuscript annotation, used hereon in) deviates by an angle of ~ 60°. Is there actually room to place a second Spz1C dimer here, assuming the same binding mode to the A chain? Does the residual poorly resolved “extra density in the original 2:1 heterodimer map” conform to the “primary binding mode”? Does reference to the “original map” mean that ResolveCryoEM removed this density? What causes the asymmetry? Is it possible to model a symmetric 2:2 complex based on the observed interfaces, or is this only possible upon significant structural rearrangements (which would be valuable information, indicating that formation of a presumably signalling competent oligomeric state comes with an energetic tradeoff)? This might provide some support for the statement “Binding of the first Spz1C homodimer to form the asymmetric complex can occur transiently at relatively low ligand concentrations but the 2:2 form would be even lower affinity, and hence only form at higher ligand concentrations”, which is otherwise currently unsubstantiated.

It is interesting that both Spz “wings” are ordered in the present complex, although these were disordered in both previous 1:1 structures (references 28 and 29). Whereas such ordering can be understood for the “distal” chain E based on contacts to the “back” and LRR14 Z-loop of chain B (Figure 1), it is not clear why this should be the case for the wing of “proximal” chain F. Are there additional contacts of the latter to chain B that would stabilise this loop? An overlay of the present Spz1C with the *Drosophila* Spz dimer (38) (Hoffmann et al., 2008) should be provided, rather than homology models (Figure S8). Indeed, I miss overlays of the present structures of the Toll5A-ECDs from the three reconstructions as well as with the full length 1:1 Toll-ECD:Spz-dimer presented in reference (29).

I would also appreciate a more thorough analysis of the binding interfaces, at least concerning the present complex and the *Drosophila* Toll-Spz complex. Is the current 2:1 structure applicable to DmToll-Spz, or would a different complex be expected? Can one expect the same overall arrangement for other Toll/Spz pairs, and can e.g. the lack of interaction between AaSpz1A and AaToll5A be rationalised in structural terms?

Concerning the AaToll5A homodimer, it is not clear whether the C-termini can sit in the same (planar?) membrane (“Such a structural arrangement –if sterically possible when the receptor is expressed on the same cell– would ensure that Toll5A is locked in an inactivate state preventing TIR domain association and signalling”) – a (supplementary) figure to this effect would be useful to the reader. Do the two chains show a strict twofold symmetry? Is there any degree of conservation (or

complementarity) of interface residues among different insect Toll species (and/or Toll-related receptors) that would suggest that this may be a dimerization mode used by several members of the Toll family in insects? In this respect, it should be noted that the authors have previously reported a low resolution of DmToll (68) (Gangloff et al., 2008) whose overall shape differs from the reconstructions presented here and tentatively interpreted in (29) as a DmToll dimer.

Other points:

Abstract: If “*A. aegypti* has evolved to become an efficient vector for arboviruses ...”, what is the selective pressure? Perhaps “*A. aegypti* is an efficient vector for arboviruses ...” would be better. A corresponding statement is also made on page 10.

p. 2, paragraph 3: Is there any evidence for the assertion “The underlying driving force of gene duplication is likely interconnected with the mosquito’s change to a hematophagous diet and the evolutionary arms-race between pathogens and insects”? How comprehensive is the database of insect Toll/Spaetzle genes (i.e. is this something specific to haematophages)?

p. 2, paragraph 4: vertebrate TLRs are unlikely to have evolved from the insect Toll9 receptor, rather a common ancestor thereof.

Can the observed antimicrobial peptide production upon exposure by Spz1C (Figure 5) be through activation of a pathway alternative to Toll5A, considering that Aag2 cells express multiple Toll forms? It would be nice to see the result of in situ activation of recombinant proSpz1C, which should be inactive but contains a TEV-protease cleavage site for activation.

Given the resolution of the SAXS data and the availability of the structures (assuming e.g. for Toll5A alone a mixture of Toll5A monomers and “cryoEM” dimers), it should be possible to fit the experimental scattering curves shown in Figure S1a. The noisy residuals shown for the analytical ultracentrifugation profiles (Figure S6) suggests that the fits are perhaps incomplete.

The “endoproteolytic degradation” of Toll5A is mentioned – and used to support the importance of the Z-loop in the interaction – but is not really explained. At what stage does this occur? How was the cleavage site detected – and is there only one cleavage site?

The link between the present data and insect viral defence/tolerance mechanisms appears to me to be somewhat tenuous – the discussion of this should either be explained more comprehensively or toned down considerably.

Minor points:

p. 2, paragraph 2: Spz proteins possess a C-terminal cystine-knot fold – add reference (38) (Hoffmann et al., 2008) in addition to (5) (Arnot et al., 2010)

p.8, first paragraph: I think “constitutively” is meant rather than “constitutionally”.

There is something wrong with the author list in reference 1.

Reviewer #2 (Remarks to the Author):

In this manuscript, Gangloff and coll. investigate the structural biology of the cytokine Spz1C and its receptor Toll5A, which are both regulated by Dengue virus infection in *Aedes* mosquitoes. Toll receptors play evolutionarily conserved roles in innate immunity from insects to mammals, hence the characterization of their structure in relation to their function in vector mosquitoes is relevant and of broad interest. The authors present the structure of Toll5A-Spz1C complexes and draw interesting comparisons with the drosophila Toll-Spz complex, which shed some light on the evolution of these important molecules at the forefront of the immune response. Most interestingly, the authors provide evidence for specific interaction between the cytokine and its receptor (there is a family of Spz cytokines and of Toll receptors) and demonstrate dose-dependent induction of antimicrobial genes by the cytokine. The article is well written, the data clearly presented, and I believe that, providing the few points listed below are addressed, this ambitious study will be of interest for a broad audience following control of arboviral infection in vector insects and molecular evolution of an important family of receptors.

1) The legend to Figure 5 is insufficient, e.g., panel C, meaning of the abbreviations for S5, DmS1, ...

- 2) About the receptor-ligand specificity: why is Spz1A used in Fig. S4 and Spz5 in Fig5C?
- 3) How does Spz5 signal induction of Gambicin? Is also through Toll5A? The authors should address the contribution of the different Toll receptors expressed in Aag2 cells on induction of GRRP and Gambicin by Spz1C and Spz5 to clarify the contribution of the contribution of these molecules.

Minor comments:

- 4) While there is solid evidence that Toll receptors function as cytokine receptors in insects, the function of Toll-7 as a pattern recognition receptor in drosophila are controversial, see PMID 27009948.

Response to Reviewer 1

- I found the study very interesting and feel that it is in principle suitable for publication in Nature Communications. Having said this, I feel that the structure(s) could/can be better described: First and foremost, I found the labelling of the chains very confusing, in particular chains B and C in the (probably most interesting) 2:1 complex – better would be B and A respectively (the latter corresponding to Toll in the previous 1:1 complexes, references 28 (Lewis et al., 2013) and 29 (Parthier et al., 2014)). Understanding would be strongly enhanced by renaming “distal” chain E (previously chain L (28) or “trailing” chain J (29)) to e.g. “D” and “proximal” chain F (M / “leading” chain K) as “P”.

We have relabelled all chains as suggested.

- In addition, the manuscript would benefit from clearer (structural) figures depicting the various interactions. As the biological relevance of the 3:1 complex is questionable, this should be moved to the Supplementary information.

We have changed Figures 1 and 2, added Supplementary Figures to depict interactions and structural comparisons. We have removed the 3:1 complex from Fig.1 as requested, while it is still described in the Supplementary information.

- In the 2:1 complex, the N-terminal region of Toll5A chain B (manuscript annotation, used hereon in) deviates by an angle of $\sim 60^\circ$. Is there actually room to place a second Spz1C dimer here, assuming the same binding mode to the A chain?

Yes, there is ample room for another Spz1C ligand, as rendered in the new **Fig.1**. There is however no clear density to build the second ligand. We modelled it based on the conformation of the ligand bound to chain A. We find that the concave binding to chain B fits the blob of density without steric clashes. There are however missing contacts at the dimerization interface, and, in particular, the Z loop is out of reach despite a fully extended Trp-loop. This is due to the asymmetric conformation adopted by the Toll5A heterodimer.

- Does the residual poorly resolved “extra density in the original 2:1 heterodimer map” conform to the “primary binding mode”? Does reference to the “original map” mean that ResolveCryoEM removed this density?

Yes, the poorly resolved “extra density” fits the “primary binding mode”. New Figure S2 shows that the density-modified maps used to build *ab initio* the atomic model of mosquito Toll5A and Spz1C, removed the extra density of the second ligand. This map can be provided upon request.

- What causes the asymmetry?

The asymmetry is most likely the result of the lack of saturation of the receptor. Residual receptor-receptor interactions remain at the N-terminal region of the receptor. Interestingly, asymmetry occurs in both the receptor and the ligand. The N-terminal cap region of the ligated receptor chain (chain A in the new nomenclature) contributes to the asymmetry at the dimer interface upon binding chain B at LRR11-13. Note that this area of receptor-receptor contacts partially overlap the homodimer interface. Receptor contacts in the homodimer are however more extensive and stretch from LRRNT1 to LRR14. Spz1C binding displaces these interactions only partially in the non-saturated and non-equilibrium snapshots visualised by cryo-EM. The LRRNT1 contact with LRR11-13 disrupts symmetrical ligand binding. The second ligand can bind at the concave side of chain B but is unable to reach the Z-loop and the junction between LRR domains of chain A (distanced by $\sim 30 \text{ \AA}$ between Trp31^{D2} and

Phe610^A instead of 3.4 Å between equivalent residues of the first ligand and chain B; and ~40 Å between Tyr85^{D2} and Ile454^A instead of 3.2Å).

There is also asymmetry in the fully resolved primary ligand. In particular, the Trp-loops of the covalent dimer that constitutes Spz1C, adopt different conformations in the proximal and distal protomers, while both beta-wings interact with the Z-loop. The proximal Trp-loop is 'curled-up' with a buried Trp31^P that caps the proximal and distal β -wings and the Z-loop, while the distal Trp-loop fully extends toward the junction between LRR domains with Trp31^D in the vicinity of Phe-610^B from LRRNT2, thus positioning the dimer juxtamembranes for near-symmetric tail-to-tail interactions.

Moreover, we modelled symmetric Spz1C molecules based on both protomers adopting the same 'curled-up' (closed) or extended (open) conformations. We find that both conformations adopt plausible geometries. The former can bind at the concave side, while the latter would clash at the N-terminal cap. Reciprocally, the latter can interact with the Z-loop and the hinge at the dimer interface. In contrast, the former with its closed β -wings and Trp-loops would collide with the Z-loop and be unable to reach the hinge region. We infer that ligand flexibility is most likely required for Spz1C to carry out receptor heterodimerization in the context of a fully saturated receptor complex in a "tug-of-war" mechanism, in which the β -wings clasp the Z-loops and the Trp-loops help orient the receptor ectodomains into position: N-termini apart and C-termini together (**Fig. R1**).

Figure R1: Towards ligand-induced symmetry: Predicted tug-of-war mechanism for double ligation of Toll receptors. In order to form a fully saturated complex, Spz Cys-knot core domain binds to the concave side of the ectodomains, its flexible loops reach the convex side of another receptor molecule and embrace its Z-loops using its β -wings. At the same time, the distal Trp-loop extends toward the dimer hinge region, and eventually brings the juxtamembranes into close proximity, while the proximal Trp-loop prevents the primary receptor from forming asymmetric receptor-receptor contacts. The receptor chains accommodate both ligands by adopting a symmetric Y-shape, with a tilt of the N-terminal LRR domains and a C-terminal crosslinking of the juxtamembrane regions.

- Is it possible to model a symmetric 2:2 complex based on the observed interfaces, or is this only possible upon significant structural rearrangements (which would be valuable information, indicating that formation of a presumably signalling competent oligomeric state comes with an energetic tradeoff)?

It is not possible to readily model the symmetric 2:2 complex as further conformational changes are needed in both the ligand and the receptor molecules. We foresee that a symmetric complex would require capping interactions at the N-terminus prone to receptor-receptor contacts (**Fig. R1**). This might be achieved by an extended open proximal Trp-loop, but it is not clear whether this region remains flexible or settles in a given conformation upon receptor contact.

In order to test this hypothesis we need to solve the structure of the symmetric 2:2 complex in excess ligand, which has eluded us so far. It is also not clear how stable such a complex would be. Parthier and al.¹ suggested that 2:2 complexes dissociate into symmetric 1:1 Toll-Spz and that another event might be required to stabilise the signalling complex. Our work raises the possibility of the Z-loop processing as the missing link in the Toll activation mechanism. Such an additional regulatory step would be reminiscent of the proteolytic activation required for nucleic-acid binding TLRs in vertebrates. Further work will be required to check this hypothesis.

In a nutshell, the switch from a signalling incompetent receptor (potentially a homodimer) to a signalling competent heterodimer upon Spz binding seem to involve the sampling of multiple conformational states to achieve disruption, relative reorientation and functional association of two receptor chains by the ligand. Such an association is governed by major entropic effects that contribute extensively to the free energy of binding. The mosquito system proved however recalcitrant to isothermal titration calorimetry analysis in contrast to *Drosophila*, probably due to the short half-lives of the receptor-ligand complexes leading to reversible ligand binding, accompanied by stoichiometric exchanges between receptor monomers and dimers in solution.

- This might provide some support for the statement “Binding of the first Spz1C homodimer to form the asymmetric complex can occur transiently at relatively low ligand concentrations but the 2:2 form would be even lower affinity, and hence only form at higher ligand concentrations”, which is otherwise currently unsubstantiated.

We have established that asymmetric 2:1 complexes form at low ligand doses in the presence of unsaturated receptors (cryo-EM, SEC-MALS, Mass Photometry data). Fully saturated receptors are thought to assemble into symmetric complexes with reciprocal receptor-ligand contacts, which exclude N-terminal receptor-receptor contacts such as those contributing to the asymmetric complex.

We therefore propose a tug-of-war mechanism of assembly for the fully saturated receptors that differs from the mechanism of asymmetric receptor assembly, in which the second ligand cannot contact the dimer interface because of the 60° deviation generated upon receptor-receptor N-terminal interactions.

Cryo-EM reveals that the concave binding interface is much smaller in the mosquito structure, compared to *Drosophila*. However, concave ligand binding seems to be an absolute prerequisite during receptor activation. The mosquito complex is short-lived in the absence of interactions at the dimer interface, which in terms of surface area is extensive. In contrast, the assembly of 2:2 complex requires the assembly of saturated receptors, which have different dynamics as suggested by AUC experiments, and remain to be characterised.

We performed mass photometry experiments at different dilutions and over different time points. We did not include the latter as binding was rapidly lost upon sample dilution (after ~5 min). In other words, mass photometry revealed the presence of labile complexes with different stoichiometries, for measurements taken within a couple of minutes. The 2:1 complex was detected at the lowest concentration in contrast to 1:1 complexes, which are end-products of heterodimer dissociation¹. In addition, AUC was measured in solution in native conditions at sample concentrations over 10-times above the overall $K_d \sim 2 \mu\text{M}$ according to our SPR

analysis using a heterogenous ligand model with two binding sites. The receptor was not saturated at equimolar ligand concentrations but required about two-fold molar excess ligand ($>70 \mu\text{M}$). It is intriguing that 2:1 complexes prevent the second ligand from reaching the dimer interface, too distant and at the wrong angle.

In the case of excess ligand, the chances of unligated receptor occurrence are slim and then, receptor-receptor interactions become rate-limiting. In the case of receptor excess and potential receptor clustering, ligand-receptor interactions become restrictive. The latter might appear counter-intuitive and the effect of avidity difficult to reconcile. The respective signalling outputs of 2:1 and 2:2 complexes are currently not known.

However, we propose based on our structural findings that this concentration dependence allows Toll receptors to produce a linear relationship between receptor occupancy and the signalling outcome at the end of the intracellular cascade.

- It is interesting that both Spz “wings” are ordered in the present complex, although these were disordered in both previous 1:1 structures (references 28 and 29). Whereas such ordering can be understood for the “distal” chain E based on contacts to the “back” and LRR14 Z-loop of chain B (Figure 1), it is not clear why this should be the case for the wing of “proximal” chain F. Are there additional contacts of the latter to chain B that would stabilise this loop?

The proximal β -wing (Spz1C Arg-72^P to Val-89^P) interacts with the proximal Trp-loop, which forms a short helix in the vicinity of the buried Trp-31^P (new **Fig.2**). The latter adopts a ‘curled-up’ conformation, in which a number of stacking interactions are capped by Phe60^A at the N-terminal region of the primary receptor chain. The proximal β -wing also interacts with its distal counterpart. The side chains of the numerous positively charged Lys residues (Lys-79, 82, 83 and 86) are poorly resolved. While Lys-79 of both protomers display centro-symmetry, Lys-83^D anchors the receptor dimer’s Z-loop Glu-456^B with a salt bridge. Another salt bridge is formed between Arg-72^D and Asp-462^B. The negatively charged Z-loop attracts Spz1C β -wings with further ionic interactions potentially stabilising a symmetrical complex upon ligand saturation. It is not clear however, how such conformational rearrangements would affect the proximal Trp-Loop. Given the flexibility of this loop, Trp-31^P exposure might ensue, which could expand interactions between Spz and the N-terminal region of the primary receptor and prevent receptor-receptor interactions as discussed above.

- An overlay of the present Spz1C with the Drosophila Spz dimer (38) (Hoffmann et al., 2008) should be provided, rather than homology models (Figure S8). Indeed, I miss overlays of the present structures of the Toll5A-ECDs from the three reconstructions as well as with the full length 1:1 Toll-ECD:Spz-dimer presented in reference (29).

Such an overlay is provided in new **Fig. S3**. Toll5A clearly has a wider N-terminal LRR domain compared to DmToll1 ($\sim 10\text{\AA}$ wider diameter). It is interesting to note that mosquito receptor chains overlay well considering their different interactions (or lack of interaction) in the homodimer, heterodimer and heterotrimer. It is not clear whether this illustrates a degree of rigidity between LRR domains or if it reflects on the limits of the techniques used. DmToll Z-loop adopts a downwards position due to crystal packing. There are mostly non conserved glycosylation sites in Toll5A: 9 are predicted but only 6 are visible, with Asn-481 and Asn-521 conserved in DmToll1 (Asn-482 and Asn-528). Toll5A Asn-521^B glycosylation interacts with Spz1C Arg-50^D-Gln-51^D at the dimer interface. Toll5A Asn-151-glycan is $\sim 10\text{\AA}$ away from the distal protomer and restricts the concave positioning of the ligand. More importantly, DmToll Asn-80-glycan prevents Spz from adopting a conformation similar to the mosquito one at the concave side upon steric clashes.

Spz1C differs from DmSpz in sequence and structure with an insertion and a deletion, compared to DmSpz, while the core of the Cys-knot overlays well. Spz1C interacts at the concave side over a smaller interface upon binding the first 7 LRRs of Toll5A chain A, compared to DmSpz, which covers the first ten LRRs of DmToll. The extended interface in DmSpz involves insertion 59-NFPQS-63. The distal β -wing in the refolded DmSpz crystal structure adopts a conformation reminiscent of Spz1C distal W-loop, which is surprising. The proximal Drosophila-style β -wing clashes at the N-ter of the primary mosquito or drosophila receptors. It is unclear whether these regions can substitute for each other in different paralogs or orthologues as their sequences are not conserved. The Trp-loops are asymmetric, with the proximal one interacting with the N-terminal region of the primary receptor and the β -wings and the distal one, extending and contacting the dimer interface. In the Drosophila Toll-Spz 1:1 complex, neither β -wings nor Trp-loops were visible indicating an entropy-driven mechanism of ligand-induced receptor dimerization.

- I would also appreciate a more thorough analysis of the binding interfaces, at least concerning the present complex and the Drosophila Toll-Spz complex. Is the current 2:1 structure applicable to DmToll-Spz, or would a different complex be expected? Can one expect the same overall arrangement for other Toll/Spz pairs, and can e.g. the lack of interaction between AaSpz1A and AaToll5A be rationalised in structural terms?

As for TLRs, the mechanism of ligand-induced dimerization may have some receptor and ligand-specific variations.

On the ligand side: Spz1C has an insertion and a deletion compared to DmSpz, located in the vicinity of the dimer interface for the former and the concave binding interface for the latter. These differences along with the changes in the peptide sequence can explain ligand specificity. AaSpz1A is much more similar to DmSpz compared to Spz1C. It is therefore not surprising that neither AaSpz1A or DmSpz bind Toll5A. However, the lack of Spz1A binding to DmToll and Toll1A remains unexplained.

On the receptor side: based on our structural analysis upon comparing the ligand binding sites in both mosquito and fly complexes, we conclude that receptor-ligand interactions are neither conserved in space or in sequence (see new **Fig. S4**). Because the mosquito complex is a transition intermediate, it is however not entirely clear whether ligand saturation would lead to a different binding mode. In an attempt to rationalise whether mosquito Toll5A could achieve a binding pause similar to Drosophila during ligand binding, receptor-ligand docking was achieved, which displayed numerous steric hindrances at the concave site. Interestingly, Drosophila Toll and Spz cannot adopt a mosquito-like binding pause because of the forbidding presence of Drosophila Asn-80-linked glycans.

- Concerning the AaToll5A homodimer, it is not clear whether the C-termini can sit in the same (planar?) membrane (“Such a structural arrangement –if sterically possible when the receptor is expressed on the same cell– would ensure that Toll5A is locked in an inactivate state preventing TIR domain association and signalling”) – a (supplementary) figure to this effect would be useful to the reader. Do the two chains show a strict twofold symmetry?

We provide additional views of the apodimer that show that the respective C-termini are compatible with the receptor molecules residing within the same plasma membrane (**Fig.1**). Given the last visible residues are 6 residues away– which can be up to 15 Å away, for a straight loop– from the predicted transmembrane helix, there is space to accommodate such an orientation, which leaves the juxtamembrane regions up to 200 Å apart. Drosophila Toll also formed homodimers², which differed in their N-terminal interactions. Lack of sequence conservation at the point of contacts suggests that these complexes are not strictly conserved.

- Is there any degree of conservation (or complementarity) of interface residues among different insect Toll species (and/or Toll-related receptors) that would suggest that this may be a dimerization mode used by several members of the Toll family in insects? In this respect, it should be noted that the authors have previously reported a low resolution of DmToll (68) (Gangloff et al., 2008) whose overall shape differs from the reconstructions presented here and tentatively interpreted in (29) as a DmToll dimer.

Unnatural concentrations or environment might trigger non physiological protein-protein contacts. Such artefacts occur in regions prone to protein-protein contacts, which cannot occur in the absence of the physiological interaction partner, and which are compensated for with a substitute. We do not exclude such a hypothesis for Toll homodimerization.

The homodimerization mode of Toll receptors does not seem to be conserved, neither is its ligand-binding mode according to the presence of Asn-80-linked glycosylation that would prevent *Drosophila* to bind its ligand in a mosquito-like heterodimer. Differences in receptor dimerization modes in the presence and absence of ligand have also been reported for vertebrate Toll-like receptors and are therefore not surprising in our point of view.

- Other points:

Abstract: If “*A. aegypti* has evolved to become an efficient vector for arboviruses ...”, what is the selective pressure? Perhaps “*A. aegypti* is an efficient vector for arboviruses ...” would be better. A corresponding statement is also made on page 10. p. 2, paragraph 3: Is there any evidence for the assertion “The underlying driving force of gene duplication is likely interconnected with the mosquito’s change to a hematophagous diet and the evolutionary arms-race between pathogens and insects”? How comprehensive is the database of insect Toll/Spaetzle genes (i.e. is this something specific to haematophages)?

There are indeed not enough insect species characterised. In general, it is believed that duplication and copy number variation can lead to changes in function. The pattern of gene gain or loss and the phylogenetic analysis of the Toll and Spz families performed by Ferreira Lima *et al.*³ indicates that functional inference might not be accurate. Characterisation of species belonging to different taxonomic groups are therefore essential. There is however evidence that Toll5A and Spz1C play a role in innate immunity in the mosquito⁴. Our work provides the transcriptional signature of Spz1C and characterises its binding to Toll5A (we have since tested and confirmed the lack of binding to Toll7 and Toll11, also expressed in unstimulated Aag2 cells along Toll1A) (**Fig. R2**). Given the strong induction of Cecropin, with proven antiviral activity, we do make a calculated leap of faith stating that the duplicated Toll5A and Spz1C might also be involved in the innate immune response against arboviruses, at least in the hemolymph in which the pathway has been shown to be functional in contrast to understudied salivary glands.

- p. 2, paragraph 4: vertebrate TLRs are unlikely to have evolved from the insect Toll9 receptor, rather a common ancestor thereof.

Apologies. The text has been changed to “Although there is a clear evolutionary relationship between insect Tolls and vertebrate TLRs the latter seem to have evolved from a common ancestor of insect Toll9 and adapted to directly recognise a diverse range of pathogen associated molecular patterns (PAMPs) such a bacterial lipopolysaccharide”.

- Can the observed antimicrobial peptide production upon exposure by Spz1C (Figure 5) be through activation of a pathway alternative to Toll5A, considering that Aag2 cells express multiple Toll forms?

We have shown that Aag2 express Toll1A, Toll5A, Toll7 and Toll11 constitutively (now Fig. S11). We have expressed their ectodomains and mixed them with 3-fold molar excess of cleaved Spz1C to find that only Toll5A was ligated under the conditions tested. However, we have not managed to express Toll10 yet and cannot rule out the presence of a Spz1C receptor that does not belong to the Toll family. Attempts at pull-down using Strep-tagged Spz1C have not been successful. Future work with Toll5A and Spz1C-specific nanobodies might be a better alternative.

Figure R2: No promiscuous pairing of Toll and Spz in *Aedes aegypti*. The ectodomains of Toll1A (T1A), Toll5A (T5A), Toll7 (T7) and Toll11 (T11) were successfully expressed in insect cells, purified and characterised for ligand binding using a range of techniques. **a** Mass Photometry; **b** Size-exclusion chromatography; **c** SDS-PAGE; and, **d** Native PAGE. Proteins in polyacrylamide gels were stained using InstantBlue® Coomassie Protein Stain (Expedeon). Ligands assessed were the cys-knot domains of Spz1C (S1C) and Spz5 (S5) upon proteolytically processing.

- It would be nice to see the result of *in situ* activation of recombinant proSpz1C, which should be inactive but contains a TEV-protease cleavage site for activation.

Fig. 5A shows the effect of recombinant proSpz1C and TEV-cleaved Spz1C on Aag2 cells. It reveals that recombinant proSpz1C is inactive in contrast to TEV-cleaved Spz1C. Aag2 cells constitutively express all members of the Spz family except Spz2 (**Fig. S11**). Kevin Maringer's group reported that Toll pathway is defective in Aag2 cells upon microbial stimulation⁵. Given that our work shows a potent production of AMP upon Spz1C addition, we hypothesise that the defect might originate upstream of Spz processing.

Attempts at *in situ* work has not been successful yet due to protein degradation upon injection in the mosquito hemolymph according to the lack of Western blot detection (anti-strep tag antibody-based). Preliminary work was performed with trypsin-cleaved Spz1C that did not have a TEV-cleavage site. The additional TEV sequence did alter Toll5A binding behaviour.

- Given the resolution of the SAXS data and the availability of the structures (assuming e.g. for Toll5A alone a mixture of Toll5A monomers and “cryoEM” dimers), it should be possible to fit the experimental scattering curves shown in Figure S1a.

We performed SEC-SAXS and SEC-MALS on the same sample preparations at 50 μ M. We are confident that there were no Toll5A monomers in the conditions tested. These experiments compare dimers of Toll5A and Spz1C-ligated Toll5A ectodomains. Next, we fitted the PDB to experimental curves as requested. The results presented below are added to the Supplemental. They show that dimers are the best fit using Crysol⁶ and Pepsi-SAXS⁷ (**Fig S9**).

PDB	.dat file	R _g (Crysol) Å	χ^2 (Crysol)	R _g (Pepsi-SAXS) Å	χ^2 (Pepsi-SAXS)
Apo-monomer	T5A	48.54	1149.9	46.9	894.36
Apo-dimer	T5A	56.65	26.35	55.3	33.22
Holo-dimer	T5A	49.37	257.47	49.2	271.5
Apo-monomer	complex	48.54	2107.23	46.9	1814.4
Apo-dimer	complex	56.81	217.68	55.3	185.2
Holo-dimer	complex	50.43	74.51	49.2	68.9

Using SREFLEX⁸ (flexible refinement of high-resolution models based on SAXS and normal mode analysis), the revised value of χ^2 (Crysol) for the apo-dimer modified by SREFLEX = 3.4. However, SREFLEX did not work with the heterodimer complex. Predictions with the program OLIGOMER⁹ were carried out to check if a combination of dimer-monomer worked better as a fit to the Toll5A data set. As suggested by SEC-SAXS, this was not the case, and the best fit remains for the dimer models.

- The noisy residuals shown for the analytical ultracentrifugation profiles (Figure S6) suggests that the fits are perhaps incomplete.

We have reprocessed the AUC data for a tighter data fit. The residual plot range below 0.05 suggests that there is a little variation of the maximal residual, which is the highest for Toll5A at high concentration. **Fig. S10** has been updated.

- The “endoproteolytic degradation” of Toll5A is mentioned – and used to support the importance of the Z-loop in the interaction – but is not really explained. At what stage does this occur? How was the cleavage site detected – and is there only one cleavage site?

A sample preparation of Toll5A ectodomain was stored at 4°C with and without Spz1C. Within a month the sample of Toll5A without ligand underwent spontaneous degradation producing a gel-shift on SDS-PAGE but not on native PAGE, while the sample with ligand remained intact. The degradation product was further characterised by in-gel mass fingerprinting to characterise the product representing the lower gel band at about 50 kDa (additional **Fig. S7**). In-gel mass fingerprinting revealed the presence of peptides from both the N-terminal and the C-terminal domain but lacked a peptide spanning the Z loop between LRR14 and LRR15 (L436-K473).

- The link between the present data and insect viral defence/tolerance mechanisms appears to me to be somewhat tenuous – the discussion of this should either be explained more comprehensively or toned down considerably.

We agree that we do not have direct evidence of the effect of these proteins on viral replication.. Our work however, reveals the transcriptional profile of Spz1C in Aag2 cells, which differs from

other Spz isoforms. Spz1C stimulates a potent production of antimicrobial Cecropin and Defensin peptides, which have been reported to have antiviral properties^{10,11}. In our hands, purified DENV NS1 protein is sufficient to repress the expression of Toll5A in Aag2 cells (our preliminary data, **Fig. S11**), while Toll5A upregulation was reported in salivary glands upon DENV2 infection¹⁰. Others have reported decreased expression of Cecropin and Defensin in wild type mosquito populations of Venezuela upon DENV1 infection¹². Hence a complicated picture emerges that needs further scrutiny.

Minor points:

- p. 2, paragraph 2: Spz proteins possess a C-terminal cystine-knot fold – add reference (38) (Hoffmann et al., 2008) in addition to (5) (Arnot et al., 2010)
- p.8, first paragraph: I think “constitutively” is meant rather than “constitutionally”. There is something wrong with the author list in reference 1.

This has now been corrected.

Reviewer #2 (Remarks to the Author):

- In this manuscript, Gangloff and coll. investigate the structural biology of the cytokine Spz1C and its receptor Toll5A, which are both regulated by Dengue virus infection in *Aedes* mosquitoes. Toll receptors play evolutionarily conserved roles in innate immunity from insects to mammals, hence the characterization of their structure in relation to their function in vector mosquitoes is relevant and of broad interest. The authors present the structure of Toll5A-Spz1C complexes and draw interesting comparisons with the *Drosophila* Toll-Spz complex, which shed some light on the evolution of these important molecules at the forefront of the immune response. Most interestingly, the authors provide evidence for specific interaction between the cytokine and its receptor (there is a family of Spz cytokines and of Toll receptors) and demonstrate dose-dependent induction of antimicrobial genes by the cytokine. The article is well written, the data clearly presented, and I believe that, providing the few points listed below are addressed, this ambitious study will be of interest for a broad audience following control of arboviral infection in vector insects and molecular evolution of an important family of receptors.
- 1) The legend to Figure 5 is insufficient, e.g., panel C, meaning of the abbreviations for S5, DmS1,

We have now changed the legend to specify that Spz5 is abbreviated by S5 and *Drosophila melanogaster* Spz1, is DmS1.

- About the receptor-ligand specificity: why is Spz1A used in Fig. S4 and Spz5 in Fig5C? 3) How does Spz5 signal induction of Gambicin? Is also through Toll5A? The authors should address the contribution of the different Toll receptors expressed in Aag2 cells on induction of GRRP and Gambicin by Spz1C and Spz5 to clarify the contribution of the contribution of these molecules.

We removed the transcriptional data on Spz1A in **Fig.5** as we do not know which receptor binds Spz1A. Spz1A did not bind Toll1A, which was unexpected as they are the closest structural homologues of *Drosophila* Toll. Similarly, Spz5 is primarily a ligand of Toll6 in *Drosophila*. Some promiscuity has been reported for Toll7¹³ and Toll11¹⁴. We have not been able to establish which receptor is responsible for Spz5 activity using purified ectodomains so far and upon testing Toll1A, Toll5A, Toll7 and Toll11, which are constitutively expressed in Aag2 cells (**Fig. S11**). Future work will aim to express and characterise the missing candidates.

The production of recombinant receptor ectodomains engineered with a cleavable Z-LRR loop will help clarify the importance of proteolytical processing of Toll receptors.

Minor comments:

- 4) While there is solid evidence that Toll receptors function as cytokine receptors in insects, the function of Toll-7 as a pattern recognition receptor in drosophila are controversial, see PMID 27009948.

Agreed and reference¹⁵ is added and the same for Toll6¹⁶.

We thank both reviewers for their time and expertise in reviewing our manuscript.

References

1. Parthier, C. *et al.* Structure of the Toll-Spatzle complex, a molecular hub in *Drosophila* development and innate immunity. *Proc. Natl. Acad. Sci. U. S. A.* **111**, 6281–6 (2014).
2. Gangloff, M. *et al.* Structural insight into the mechanism of activation of the toll receptor by the dimeric ligand Spätzle. *J. Biol. Chem.* **283**, (2008).
3. Lima, L. F., Torres, A. Q., Jardim, R., Mesquita, R. D. & Schama, R. Evolution of Toll, Spatzle and MyD88 in insects: the problem of the Diptera bias. *BMC Genomics* **22**, (2021).
4. Shin, S. W., Bian, G. & Raikhel, A. S. A toll receptor and a cytokine, Toll5A and Spz1C, are involved in toll antifungal immune signaling in the mosquito *Aedes aegypti*. *J. Biol. Chem.* **281**, 39388–39395 (2006).
5. Russell, T. A., Ayaz, A., Davidson, A. D., Fernandez-Sesma, A. & Maringer, K. Imd pathway-specific immune assays reveal nf-kb stimulation by viral rna pamps in *Aedes aegypti* aag2 cells. *PLoS Negl. Trop. Dis.* **15**, (2021).
6. Franke, D. *et al.* ATSAS 2.8: A comprehensive data analysis suite for small-angle scattering from macromolecular solutions. *J. Appl. Crystallogr.* (2017). doi:10.1107/S1600576717007786
7. Grudin, S., Garkavenko, M. & Kazennov, A. Pepsi-SAXS: An adaptive method for rapid and accurate computation of small-angle X-ray scattering profiles. *Acta Crystallogr. Sect. D Struct. Biol.* **73**, (2017).
8. Panjkovich, A. & Svergun, D. I. Deciphering conformational transitions of proteins by small angle X-ray scattering and normal mode analysis. *Phys. Chem. Chem. Phys.* **18**, (2016).
9. Konarev, P. V., Volkov, V. V., Sokolova, A. V., Koch, M. H. J. & Svergun, D. I. PRIMUS: A Windows PC-based system for small-angle scattering data analysis. *J. Appl. Crystallogr.* (2003). doi:10.1107/S0021889803012779
10. Luplertlop, N. *et al.* Induction of a peptide with activity against a broad spectrum of pathogens in the *Aedes aegypti* salivary gland, following infection with Dengue Virus. *PLoS Pathog.* **7**, e1001252 (2011).
11. Mukherjee, D., Das, S., Begum, F., Mal, S. & Ray, U. The mosquito immune system and the life of dengue virus: What we know and do not know. *Pathogens* **8**, (2019).
12. Méndez, Y., Pacheco, C. & Herrera, F. Inhibition of defensin and cecropin responses to dengue virus 1 infection in *Aedes aegypti*. *Biomedica* **41**, (2020).
13. McIlroy, G. *et al.* SI - Toll-6 and Toll-7 function as neurotrophin receptors in the *Drosophila melanogaster* CNS. *Nat. Neurosci.* **16**, 1248–56 (2013).
14. Nonaka, S. *et al.* Characterization of Spz5 as a novel ligand for *Drosophila* Toll-1 receptor. *Biochem. Biophys. Res. Commun.* (2018). doi:10.1016/j.bbrc.2018.10.096
15. Lamiable, O. *et al.* Analysis of the contribution of hemocytes and autophagy to *Drosophila* antiviral immunity. *J. Virol.* JVI.00238-16 (2016). doi:10.1128/JVI.00238-16
16. Angleró-Rodríguez, Y. I., Tikhe, C. V., Kang, S. & Dimopoulos, G. *Aedes aegypti* Toll pathway is induced through dsRNA sensing in endosomes. *Dev. Comp. Immunol.* **122**, (2021).

REVIEWERS' COMMENTS

Reviewer #1 (Remarks to the Author):

The revised manuscript is considerably improved and to my mind is suitable for publication in Nature Communications.

My only remaining criticism concerns the Z-loop cleavage (Figures S6 and S7), which is still not described experimentally. Under what conditions was the cleaved Toll5A ECD (T5A*) found? Was this detected e.g. after prolonged storage (if so, how long, what temperature), or was a cleaved component already present following purification whose amount increased with time? Is Toll5a susceptible to asparagine endopeptidase (AEP) cleavage? If so, do AEP inhibitors interfere with the appearance of T5A*?

I think that at the very least the experimental details for the appearance of T5A* should be provided, as the authors devote the penultimate paragraph of the discussion to a putative role of Z-loop processing in Toll signaling. In this respect, they should delete the word "also" from the final sentence "Of note, the nucleic-acid binding TLRs that are also activated by endoproteolytic cleavage display positive cooperativity".

Reviewer #1 (Remarks to the Author):

“The revised manuscript is considerably improved and to my mind is suitable for publication in Nature Communications.

My only remaining criticism concerns the Z-loop cleavage (Figures S6 and S7), which is still not described experimentally. Under what conditions was the cleaved Toll5A ECD (T5A*) found? Was this detected e.g. after prolonged storage (if so, how long, what temperature), or was a cleaved component already present following purification whose amount increased with time? Is Toll5a susceptible to asparagine endopeptidase (AEP) cleavage? If so, do AEP inhibitors interfere with the appearance of T5A*?

I think that at the very least the experimental details for the appearance of T5A* should be provided, as the authors devote the penultimate paragraph of the discussion to a putative role of Z-loop processing in Toll signaling. In this respect, they should delete the word "also" from the final sentence "Of note, the nucleic-acid binding TLRs that are also activated by endoproteolytic cleavage display positive cooperativity".”

Apologies for the lack of details. The first occurrence of Toll5A ECD sample degradation was after AUC in the absence of Spz1C and long-term storage in the fridge as follows: upon AUC, samples of Toll5A ECD and Toll5A ECD combined with S1C ligand were recovered, placed in Eppendorf tubes and stored at 4°C. After 3 weeks, the samples were re-analysed by SDS-PAGE and native PAGE. Toll5A was intact in the presence of Spz1C but was cleaved in half with both N-terminal and C-terminal regions intact in the absence of Spz1C. Mass spec confirmed the cleavage occurred within the loop located between LRR14 and LRR15.

Next, limited proteolysis using asparagine endopeptidase AspN (Endoproteinase AspN, purified from *Flavobacterium meningosepticum*, by NEB, Cat. No. P8104S) was attempted confirming that Toll5A ECD is susceptible to AEP processing (**Fig.R1**). However, bacterial AspN failed to produce a clean endoproteolytical cut. Although a protein band at ~ 50kDa was observed, further degradations were also present, suggesting a lack of specificity of this enzyme.

We have not tested the effect of AEP inhibitors on Toll5A as there is no evidence of spontaneous cleavage in mosquito cell lines Aag2 and C6/36 (**Fig.R2**). Interestingly, Toll1A full ECD was not produced in these cell lines in contrast to Sf9 cells. Western blot revealed that Toll1A was expressed as a 50kDa truncation. The implications of these findings remain to be further characterised.

Figure R1: Toll5A ectodomain is susceptible to asparagine endopeptidase cleavage.

a Time course of Toll5A proteolysis in the presence of AspN. **b** Processing seems to occur reproducibly within a couple of hours at 37°C. **c** The presence of the ligand Spaeztle1C (S1C) affects the AspN cleavage pattern. However, further degradations occur within the ectodomain of Toll5A using AspN that the ligand does not protect. Typical 4-20% SDS-PAGE gels are shown, with protein stained with InstantBlue® Coomassie Protein Stain (Abcam, Cat. No. ISB1L). **d** Predicted AspN cleavage sites within Toll5A LRR14.

Figure R2: Toll5A is expressed as a full-length ectodomain in contrast to Toll1A.

Recombinant baculoviruses encoding the ectodomains of Toll1A (~ 120 kDa) and Toll5A (~ 110 kDa) followed by a C-terminal FLAG-tag fusion peptide have been used to infect mosquito cell lines Aag2 (a), and C6/36 (b); and insect cells Sf9 (c-d). Proteins in culture supernatants were analysed by Western blot over time using anti-Flag tag primary antibody (Sigma-Aldrich Mouse Anti-FLAG® M2, Cat. No. F3165-2MG) and anti-mouse IgG-HRP secondary antibodies (Sigma-Aldrich Goat Anti-Mouse IgG-Peroxidase, Cat. No. A4416-1ML). In contrast to full length Toll5A, Toll1A ECD appears as a weak full-length apparent band at 150 kDa, while a truncation at ~50 kDa prevails.